# Direct atomic-scale investigation of the coarsening mechanisms of exsolved catalytic Ni nanoparticles

Dylan Jennings[1,2,3,9] ✉, Moritz L. Weber[4,10,11] ✉, Ansgar Meise[2], Tobias Binninger[5], Conor J. Price[5], Moritz Kindelmann[1,2,6,12], Ivar Reimanis[7], Hiroaki Matsumoto[8], Pengfei Cao[2], Regina Dittmann[4], Piotr M. Kowalski[5], Marc Heggen[2], Olivier Guillon[1], Joachim Mayer[2,6], Felix Gunkel[4] ✉ & Wolfgang Rheinheimer[3] ✉

Exsolution-active catalysts allow for the formation of highly active metallic nanoparticles, yet recent work has shown that their long-term thermal stability remains a challenge. In this work, the dynamics of exsolved Ni nanoparticles are probed in-situ with atomically resolved secondary electron imaging with environmental scanning transmission electron microscopy. Pre-characterization shows embedded $NiO_x$ nanostructures within the parent oxide. Subsequent in-situ exsolution demonstrates that two populations of exsolved particles form with distinct metal-support interactions and coarsening behaviors. Nanoparticles which precipitate above embedded nanostructures are observed to be more stable, and are prevented from migrating on the surface of the support. Nanoparticle migration which fits random-walk kinetics is observed, and particle behavior is shown to be analogous to a classical wetting model. Additionally, DFT calculations indicate that particle motion is facilitated by the support oxide. Ostwald ripening processes are visualized simultaneously to migration, including particle redissolution and particle ripening.

The development of highly active, nanostructured composite materials for heterogeneous catalysis is a critical component of a successful transition away from carbon-intensive energy sources[1]. To improve the design of oxide-supported metal catalysts, attaining a mechanistic understanding of modes of catalyst degradation is pivotal[2,3]. One of the major deactivation pathways of metal nanoparticle catalysts is a thermally-driven decrease in the density of catalytically active triple-phase-boundaries, through nanoparticle coarsening[3]. Metal exsolution provides a promising route for the production of oxide-supported catalytic nanoparticles for a wide variety of potential catalytic

[1]Materials Synthesis and Processing (IMD-2), Institute of Energy Materials and Devices, Forschungszentrum Jülich GmbH, Jülich, Germany. [2]Ernst Ruska-Centre for Microscopy and Spectroscopy with Electrons (ER-C), Forschungszentrum Jülich GmbH, Jülich, Germany. [3]Institute for Ceramic Materials and Technologies, University of Stuttgart, Stuttgart, Germany. [4]Electronic Materials (PGI-7), Peter Grünberg Institute, Forschungszentrum Jülich GmbH, Jülich, Germany. [5]Theory and Computation of Energy Materials (IET-3), Institute of Energy Technologies, Forschungszentrum Jülich GmbH, Jülich, Germany. [6]Central Facility for Electron Microscopy (GFE), RWTH Aachen University, Aachen, Germany. [7]Department of Metallurgical and Materials Engineering, Colorado School of Mines, Golden, CO, USA. [8]Hitachi High-Tech Corporation, Core Technology & Solution Business Group, Ibaraki, Japan. [9]Present address: Advanced Transmission Electron Microscopy, Faculty of Physics and Astronomy, Ruhr-University-Bochum, Bochum, Germany. [10]Present address: Next-Generation Fuel Cell Research Center, Kyushu University, Fukuoka, Japan. [11]Present address: Department of Materials Science and Engineering, Massachusetts Institute of Technology, Cambridge, MA, USA. [12]Present address: DTU Energy, Technical University of Denmark, Lyngby, Denmark. ✉e-mail: dylan.jennings@ruhr-uni-bochum.de; weber.lukas.moritz.320@m.kyushu-u.ac.jp; f.gunkel@fz-juelich.de; wolfgang.rheinheimer@ifkb.uni-stuttgart.de

applications (e.g., steam or $CO_2$ electrolysis, steam methane reforming, oxygen reduction reaction, and others)[4–7]. In the exsolution process, the redox-stable parent oxide is doped with reducible cations of the catalytically active metal species. Upon reduction, a precipitation reaction occurs, resulting in the formation of highly dispersed metallic nanoparticles on the oxide surface. Many benefits of exsolution catalysts have been demonstrated, including ease of preparation of homogeneously distributed catalytic nanoparticles[8], improved activity[9], and increased nanoparticle stability[10]. In particular, the improved thermal stability of exsolved nanoparticles as opposed to those prepared by conventional methods is significant. However, recent work has clearly shown that there are still limitations to the thermal stability of exsolved nanoparticles[11], particularly for acceptor-doped perovskites, which are relevant for energy conversion devices that require fast oxygen exchange kinetics. As such, it is important to obtain a fundamental understanding of the mechanisms that can impact nanoparticle stability in exsolution-active systems.

A multitude of factors have been posited to impact the formation and stability of exsolved nanoparticles; these include 'socket' formation[10], support orientation[12,13], strain effects[14,15], and the defect chemistry of the support surface (including surface space charge)[5,11,16]. For all oxide-supported metal catalysts, a strongly bonded interface between the nanoparticles and the support is beneficial to stability. In the case of reducible oxide supports, a reversible encapsulation of the particle by the support species can occur in reducing conditions, which (when combined with a loss in chemisorption) is referred to as a 'strong metal-support interaction' (SMSI)[17–19]. Nanoparticle stability is a thoroughly-studied topic in the literature, where it is well accepted that particle growth can occur via two routes: Ostwald ripening (OR) and particle migration and coalescence (PMC)[20–24]. In OR, coarsening occurs by the diffusion of atomic species from small to large particles, whereas in PMC, particles are themselves mobile in the system and grow through sintering upon physical contact with another particle. Note that while the coarsening of supported particles differs from classical OR, the term has been extended to cover this process[22]. The dominant coarsening mechanism is expected to be influenced by several factors, including particle size/size-distribution, temperature, atmosphere, and support defect chemistry[11,20,23].

To elucidate the true behavior of catalytic systems, in-situ investigations are often required, with transmission electron microscopy (TEM) frequently selected as the characterization method of choice[25–28]. In prior work, surface-sensitive atomic-resolution imaging of oxides has been shown to be possible by using a probe-corrected scanning TEM (STEM) equipped with a secondary electron (SE) detector[29]. However, the capability for atomic-resolution secondary electron imaging has yet to be demonstrated during in-situ STEM experiments at elevated temperatures. In the current work, the combination of a plan-view liftout procedure onto a MEMS chip and a probe-corrected environmental STEM equipped with an SE detector is deployed; the setup allows for in-situ atomic-resolution secondary electron imaging of catalytic nanoparticles supported on an aligned, flat, single-crystalline oxide surface. The model experiment enables observation of the fundamental behavior of exsolved metal particles.

For this study, a thin film of Ni-doped strontium titanate ($SrTi_{0.95}Ni_{0.05}O_{3-\delta}$, referred to as STNi) has been epitaxially grown by reflection high-energy electron diffraction (RHEED) controlled pulsed laser deposition (PLD) (Supplementary Fig. 1) to act as a model system for an exsolution catalyst. In the STNi system, the exsolution of Ni nanoparticles can be achieved[11], and a morphology consisting of $NiO_x$ nanostructures coherently embedded inside the layer has been shown in a similar system[30], which may or may not assist the exsolution of NPs. Nanoscale ordering of dopant species is common in doped perovskites[31–33], and embedded nanoparticles are expected to precipitate in the material bulk during exsolution[9,34]. Hence, interactions between enriched dopants and nanoparticles in the subsurface with

the catalytically accessible surface nanoparticles can be expected for many exsolution-type catalysts. As a result, the STNi thin film is an ideal model system for understanding the interplay between exsolved particles and the defect structure of the underlying oxide support.

To begin, the film is characterized ex-situ by STEM, displaying the presence of heteroepitaxially embedded nanostructures of $NiO_x$ inside the Ni-doped strontium titanate film. Next, the experimental setup for in-situ STEM experiments is presented and demonstrated to allow for atomic-resolution SE imaging at elevated temperatures and reducing atmospheres. The combination of SE and high-angle annular dark-field (HAADF) imaging modes is revealed to be ideal for understanding the impact of the underlying support structure on the behavior of the supported Ni nanoparticles. Two populations of Ni nanoparticles are shown to form, which display different coarsening behaviors. Fundamental insights into particle migration and OR are gained through in-situ observations, which can be broadly applied to better understand thermal deactivation processes in oxide-supported metal catalysts.

## Results
### Thin film characterization prior to exsolution
Epitaxial thin films of STNi were grown by PLD, monitored by reflection high-energy electron diffraction (RHEED), to a thickness of 150 nm. XRD of the films indicates high crystallinity, while the RHEED surface pattern and AFM results show that a smooth surface for as-grown films is present (Supplementary Fig. 1). A plan-view liftout procedure was used to access a large {001} surface for in-situ measurements. Figure 1 summarizes the as-grown defect structure of the STNi thin film in the plan-view using HAADF imaging and energy dispersive X-ray spectroscopy (EDS), prior to any in-situ experiments. All imaging in Fig. 1 is done with a HAADF detector (providing detail on the bulk structure of the film).

The plan-view imaging shows a web-like structure of bright contrast (Fig. 1a, b), despite a fully single-crystalline film, which corresponds to regions of higher Ni content; when viewing the cross-section of the same film (Supplementary Fig. 2), the boundaries of the web-structure are observed to penetrate through the full thickness of the film. At high resolution, it is clear that second-phase nanocolumns of $NiO_x$ are coherently embedded within the Ni-doped strontium titanate matrix. The embedded $NiO_x$ nanocolumns are not on the surface of the sample, but are instead fully surrounded by the lattice of the thin film (Supplementary Fig. 3), and are observed to have nucleated near the triple junctions of the Ni-rich boundaries. The structures are fully revealed through the utilization of plan-view imaging, whereas they are obscured by the lamella thickness in the cross-section view (Supplementary Fig. 2), as is typically done in exsolution studies in the literature. The features in Fig. 1 and Supplementary Fig. 2 are consistent with cross-section analyses of thin films with the same or similar compositions[11,30]. Further characterization of the as-grown film is shown in Supplementary Figs. 2, 6–9.

Thermodynamic modeling was performed to understand the solubility of Ni in $SrTiO_3$ by calculating the Margules interaction parameter, $W$, with several approaches, including density functional theory (DFT). Calculations indicate that the solubility limit of Ni in $SrTiO_3$ is below the Ni content in the as-synthesized composition (see Supplementary Table 2) at the conditions of deposition. As a result, secondary phases would be expected to form, fitting with the observation of $NiO_x$ nanostructures in the as-grown film (Fig. 1).

### Exsolution behavior of Ni nanoparticles
To understand the impact of embedded nanostructures on the behavior of exsolved particles, it is necessary to view both the sample surface and the bulk structure with high spatial resolution. Using the current experimental design (see Supplementary Fig. 10 for a schematic view), exsolved Ni nanoparticles can be imaged on the surface of the lamella by atomic-resolution SE imaging, and the support defect

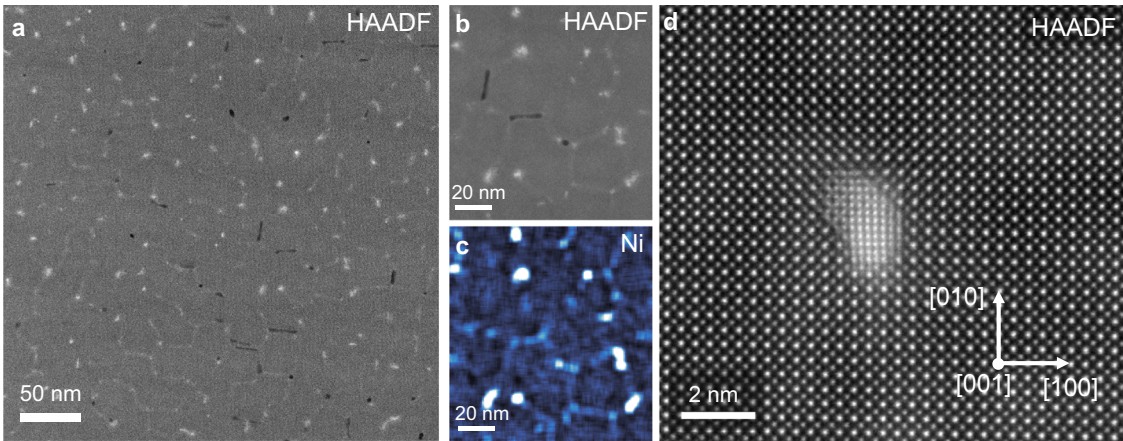

**Fig. 1 | Ex-situ plan-view characterization of the as-grown Ni-doped SrTiO₃ thin film. a, b** High-angle annular dark-field images, showing a web-pattern of bright contrast through the single-crystal film. **c** EDS mapping from the region in (**b**) shows that bright contrast corresponds to regions of increased Ni content. **d** High-resolution STEM-HAADF imaging reveals embedded NiO$_x$ nanostructures within a Ni-doped SrTiO$_3$ matrix.

structure can be simultaneously observed using conventional transmission (e.g., HAADF) detectors. In-situ exsolution experiments were conducted using pure hydrogen, with pressures in the column of ≈1 Pa. Samples were systematically heated, starting at 300 °C and increasing in temperature by 50 °C every 15 min until particles were observed to precipitate. In a close match to previous ex-situ studies of the same system[11,16], the nanoparticles were first observed to exsolve at 400 °C. As a confirmation that the in-situ experimental setup reproduces ex-situ results, the kinetics of Ni particle growth demonstrates qualitatively similar behavior as prior literature from the same system[11] (see Supplementary Fig. 11).

As demonstrated in Fig. 2, atomic-resolution secondary electron imaging at an elevated temperature (here, 500 °C) is feasible, allowing for detailed characterization of Ni nanoparticles that have exsolved on the surface of the plan-view lamella. The combination of surface-sensitive secondary electron imaging and bulk-sensitive HAADF imaging allows for correlation between particle location and morphology with support structure. Atomic-resolution SE imaging at room temperature has been demonstrated to be possible by several groups previously[29,35–37]. The contrast in atomic-resolution SE imaging arises from inner-shell ionization events triggered by the primary electron beam, and prior literature has indicated that SE contrast is less sensitive to specimen tilt off-zone than other imaging modes[36]. The nanoparticles can be divided into two different populations, based on their metal-support interaction: some Ni particles were observed to precipitate above a NiO$_x$ nanocolumn in the thin film, and others nucleated on 'pristine' areas of the Ni-doped strontium titanate matrix (Fig. 2). From this point onward, these two particle populations will be referred to as 'nanocolumn-associated', and 'pristine'. The presence of two particle populations proves that, in addition to the Ni-STO matrix, particle precipitation from embedded nanostructures also contributes to the exsolution characteristics of the material system. While annealing the particles in the microscope (at temperatures ranging from 400 °C, when exsolution was observed, to 500 °C), the ratio of the two particles changes significantly. While the density of nanocolumn-associated particles stays constant, the density of pristine particles drops precipitously (Fig. 2h). Both nanoparticle types are shown to precipitate in close proximity to one another (Fig. 2a), and nanocolumn-associated particles are larger on average (Fig. 2i). A calculation of exsolved Ni volume (Supplementary Fig. 4) shows that the overall Ni volume increases during the experiment, indicating that exsolution occurs simultaneously with nanoparticle coarsening. The HAADF images in Fig. 2a, b show that the nanocolumns are present in the film after exsolution has occurred, though it is expected that they

also reduce to metallic Ni during the experiment[30]. Similarly, embedded metal nanoparticles have been shown to nucleate from solid-solution parent oxides[9,38] within the oxide matrix.

Due to the high sensitivity of SE imaging to the sample surface[29,37], the crystal lattice of the larger particles (e.g., Fig. 2b–d) can be detected, fitting to that of a [100] oriented Ni particle with a 'cube-on-cube' heteroepitaxial relationship to the STNi support. The surface facets of the particles can also be clearly seen, with {100} and {110} edges being visible from the top-down viewing direction (refer to Supplementary Fig. 12 for additional particle images, including edge annotations). Figure 2b displays a high-resolution SE image of a relatively large particle (≈5 nm diameter), which demonstrates the capability of the combination of the sample preparation and microscope setup utilized for in-situ experiments. The contrast in the particle in Fig. 2b is brightest in the center, and becomes darker near the edge – a clear indication of its 3-dimensional faceted morphology. Additional surface features are evident through SE imaging, such as the visibility of a surface step in the right-most (100) facet of the particle. Taken together, the images in Fig. 2 demonstrate the high potential for atomically resolved secondary electron imaging of oxide-supported metal nanoparticles for both in-situ and ex-situ analysis.

### Dynamics of exsolved particles: particle migration

Despite previous studies on the migration of oxide-supported metal nanoparticles[20,25], the underlying mechanisms remain unclear. A common explanation of particle migration in such systems is that atomic diffusion on the surface of a metal nanoparticle causes a change in the center of mass of the particle, resulting in a Brownian-motion type diffusion of the particle as a whole[39,40]. It has been speculated that above the Tammann temperature (≈$0.5T_m$), particles can exist in a quasi-liquid state[41], yet the mechanism and the external gradients that influence particle migration are still not well understood, particularly for oxide-supported systems. Enhanced knowledge of the mechanisms of particle migration is a critical first step towards designing materials that are resistant to coarsening via PMC.

A particular benefit of the present experimental design is the ability to visualize particles on a flat surface, allowing for quantitative investigations of particle migration. Surface curvature, which is prevalent in prior in-situ TEM experiments of catalytic nanoparticle dynamics[20,25,42–44], would be expected to impact particle mobility by influencing capillary forces, and makes it nearly impossible to track particle movement (given that particles will move along the viewing direction). In Fig. 2, it is shown that exsolution from the STNi thin films allows for the formation of two distinct nanoparticle populations.

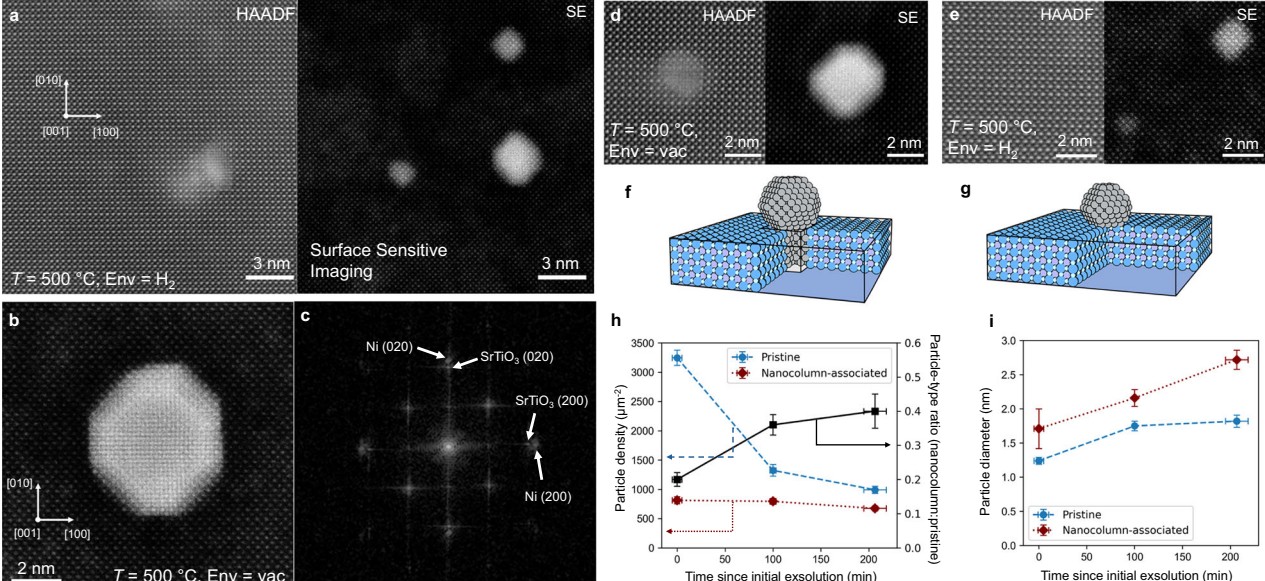

**Fig. 2 | Formation of two distinct nanoparticle populations during in-situ exsolution.** Two different particle morphologies are shown to form in situ: nanocolumn-associated and pristine. **a** Atomically resolved HAADF and SE images of a region with three exsolved particles visible, with two pristine and one nanocolumn-associated particle. An SE image of a relatively large particle (**b**) shows faceting, surface steps, and the Ni crystal lattice. FFT analysis in (**c**) shows a 'cube-on-cube' epitaxial relationship, with lattice parameters which fit with expected: $SrTiO_3 \approx 3.9$ Å, $Ni \approx 3.5$ Å. Higher resolution HAADF and SE images of the two particle types are shown in (**d**) and (**e**). Crystal models of the two different particle populations are shown in (**f**, **g**). Color code: Ni is gray, Sr is purple, Ti is dark blue, and O is light blue. Disparities in both particle density and size are observed over the course of the in-situ experiments (**h**, **i**). All images were collected at 500 °C, either in vacuum (**b**, **d**) or an atmosphere of $\approx 1$ Pa hydrogen (**a**, **e**). The data in (**h**, **i**) was collected during experiments with temperatures beginning from 400 °C and reaching up to 600 °C, in hydrogen for $\approx 100$ min, then in vacuum for $\approx 100$ min. The exact temperature ramps and conditions are displayed in Supplementary Fig. 5. An average of 46 particles per timestep were evaluated for the data in (**h**, **i**), and error bars represent the standard error of the mean of the measurement.

Given that metal-support interactions are critical to catalytic stability[17,45], it is unclear how the variant interfaces of the nanocolumn-associated and pristine particles will impact particle migration behavior. Metal-support interactions are expected to impact the PMC process, as particle migration should be inhibited by a more stable interface arrangement. During the in-situ experiments, particle migration was observed in multiple cases, with examples shown in Fig. 3. Three particles are visible in Fig. 3a–c, as revealed by simultaneous SE and HAADF imaging: two nanocolumn-associated, and one pristine nanoparticle, allowing for a direct comparison of the migration behavior of the two particle types in close proximity. Only the pristine particle is observed to migrate on the surface, suggesting that the nanocolumn-associated particles are being stabilized, or pinned, by the second-phase nanocolumns in the film. Several videos showing particle movement were captured, with each showing movement only of pristine nanoparticles (refer to supplementary Movies 1 and 2 to observe the particle movement in Fig. 3a–c). For additional evidence of particle movement in pristine nanoparticles, refer to supplementary Movies 3, 5, and 6.

Given that imaging is done perpendicular to a flat surface, the two-dimensional particle movement of the mobile nanoparticle can be tracked and fit to expected kinetic models. The particle was tracked for 5 min at 500 °C (Fig. 3d), displaying kinetics which fit excellently to a random-walk diffusion model (Fig. 3e, measured diffusion coefficient $D_{NP} = 0.4$ Å$^2$ s$^{-1}$). The high-quality fit of the model in Fig. 3e suggests that the particle movement, which leads to PMC, is a random-walk diffusion process when moving on a flat surface. The analysis in Fig. 3e presents a direct measurement of random-walk kinetics for an oxide-supported metal nanoparticle, an analysis made possible by the in-situ experimental design. The Brownian-motion type process is predicted by the classical view of migration as a result of the mobility of atoms on the surface of the nanoparticle[39], but could be recreated by other mechanisms of movement.

Particle mobility of several particles at higher temperatures (700 °C) was also observed over the course of two minutes of imaging (Fig. 3f, g). The diffusivities of each pristine particle (49 total) were tracked over the time period, showing a skewed distribution with $\approx 10$ particles that have significant mobility, while the majority were relatively stable. The ensemble diffusivity at 700 °C in vacuum was measured to be $D_{NP} = 1.0$ Å$^2$ s$^{-1}$. While limited data was taken to evaluate the temperature-dependence, comparison of the diffusivity at 500 °C and 700 °C would reflect an activation energy of $\approx 0.6$ eV, by assuming that the nanoparticle diffusion process follows a typical Arrhenius relationship (Eq. (3)). The evidence in Fig. 3 demonstrates that particle migration can occur even when no obvious driving force (i.e., curvature, external field, etc.) is present. Exsolved particles are well known to be 'socketed' to the support oxide[10], but no indication of socketing is observed here, fitting with ex-situ analysis of a closely related system[11].

The video of the particle movement (Supplementary Movies 1, 2) shows that particle migration occurs through a series of larger jumps, followed by periods of relative stability. The larger jumps match closely with the {100} type crystallographic directions of the support, suggesting that the jumps that combine to form random-walk-type diffusion kinetics may be more favorable along certain crystallographic directions. The particle morphology is displayed in Fig. 4a–c for one of the largest jumps, where the particle is observed to elongate to facilitate its movement; in other words, the particle changes shape from circular to oblong, to circular again after the jump. The variation in particle shape implies a more complex process than the conventional view of particle migration resulting from a series of atomic displacements on the particle surface, suggesting instead that particle movement dynamics can be described as a wetting phenomenon. A similar shape change can be modeled using a classical wetting physics model of a droplet on a support with spatial inhomogeneities in wettability (i.e., the equilibrium contact angle) (Fig. 4d–f). Matching with this view, it is understood that wetted droplets can change shape

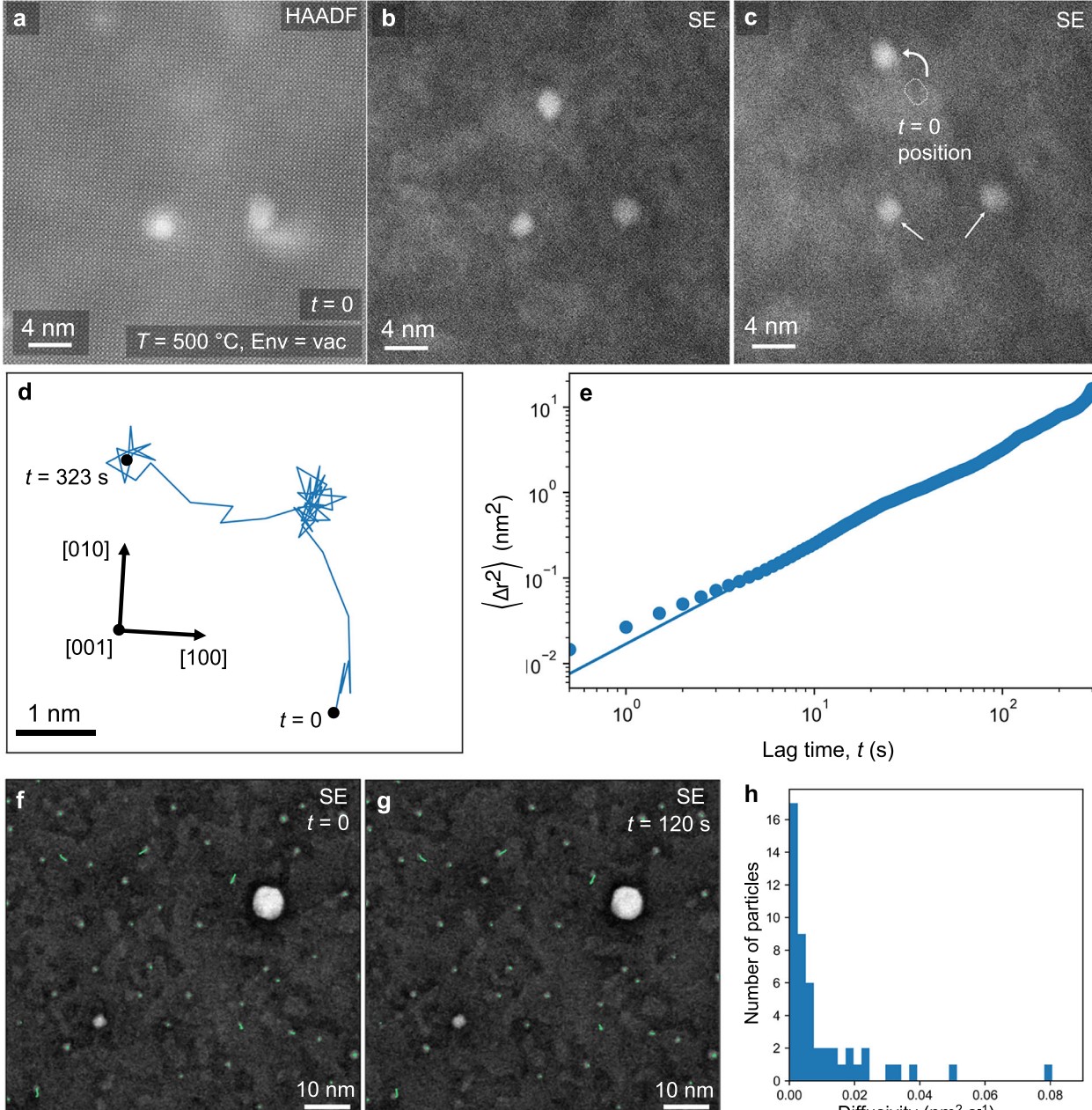

**Fig. 3 | Kinetics of nanoparticle migration.** Exsolved Ni particle movement was observed in STNi at multiple temperatures in vacuum by SE imaging. **a–c** HAADF and SE imaging of the region of interest before and after 310 s, showing that the pristine particle has migrated at 500 °C. **d** the track of the particle throughout the entire 5-min video, and **e** plotting of particle mean squared displacement as a function of time, along with a power law fit (solid line). Two images showing migration at 700 °C (initial (**f**) and final state (**g**)) with particle tracks (green lines) overlaid on all pristine particles. The diffusivities of all particles were calculated (**h**), showing that ≈20% (10/49) of the pristine particles are observed to migrate significantly. Data in (**a–c**) was collected at 500 °C in vacuum, and data in (**f, g**) was collected at 700 °C in vacuum.

as a result of the inhomogeneities on the support surface, which cause fluctuations in capillary forces[46]. The experimental observation in Fig. 4a–c was at 500 °C, just below the Tammann temperature of Ni (≈600 °C)[47], the temperature above which a supported metal particle is expected to behave similarly to a liquid droplet. In the case of an oxide-supported metal particle, local fluctuations in the chemistry (for example, cation stoichiometry and oxygen vacancy concentration) of the surface of the oxide support are a potential cause of the changes in capillary forces.

DFT modeling was performed to better understand the behavior of the nanoparticle during large jumps (Fig. 4g–i). Static DFT simulations were performed of a nanoparticle in an initial (pre-jump) and in an elongated shape (mid-jump), made to model the experimental conditions as closely as possible (that is, the size and morphologies observed in Fig. 4a, b combined with the faceting and orientation observed in HRSE imaging in Fig. 2). Given that particle strain would need to be greater than 30% to recreate the elongation in Fig. 4b, atoms were assumed to rearrange between the two states, keeping the total number of 409 atoms fixed. In each state, the particle was modeled both in isolation and attached to the support oxide, providing detailed information about the influence of the support on the particle shape changes. The support was modeled as a

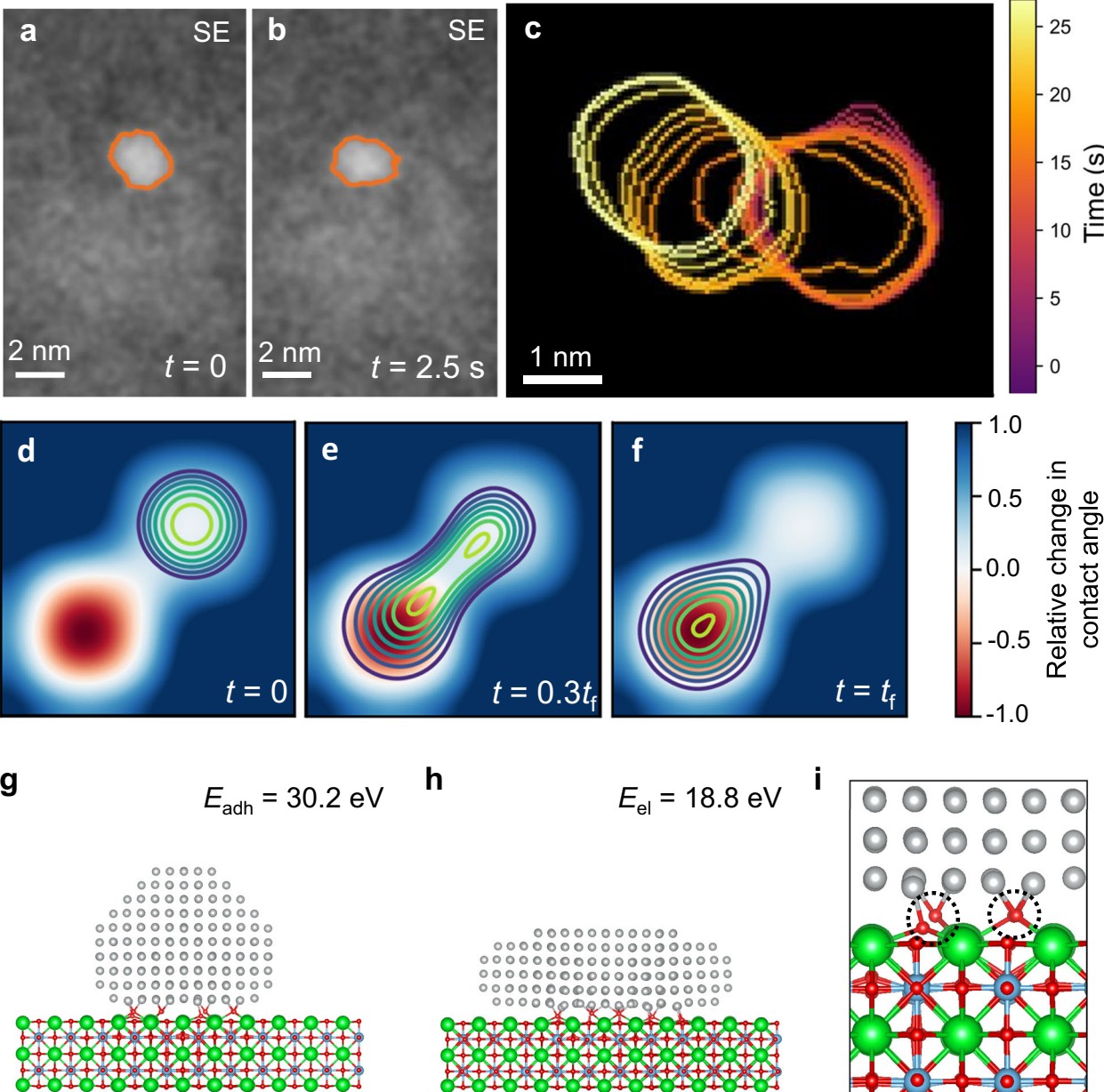

**Fig. 4 | Mechanism of nanoparticle migration.** Shape changes in the particles were observed during movement, which can be associated to wetting phenomena. **a**, **b** Two frames from a single large jump the particle made, during which a shape change is observed (outlining of the particle was done automatically by image thresholding). **c** An overlay of a series of particle shapes over the ≈25 s during which the particle jump in (**a**, **b**) occurs, with color differences indicating temporal variation. (**d**–**f**) Classical wetting physics simulation of a liquid droplet on a support with heterogeneous wettability, showing that particle shape changes can be represented as a wetting phenomenon. DFT modeling of a static particle in the 'initial' (**g**) and 'elongated' (**h**) states. DFT modeling indicates that the increase in energy corresponding to particle elongation ($E_{el}$) is lowered by its interaction with the support. **i** Zoom of the particle:support interface in (**h**), showing that bonding between the particle and support is enhanced by the displacement of oxygen towards the Ni atoms. Color code: green = Sr, blue = Ti, red = O, gray = Ni. The images in (**a**–**c**) were collected at 500 °C in vacuum.

stoichiometric $SrTiO_3$ slab with a SrO terminating layer, as observed in HAADF images of the edge of the experimental sample (Supplementary Fig. 15). Hu and Li[48] have developed a thermodynamic descriptor for the expected onset temperature of PMC, based on an interfacial area-normalized adhesion energy, defined as adhesion energy ($E_{adh}$) multiplied by the number of atoms per interfacial area ($S^m$). The adhesion energy of the symmetrical particle ($E_{adh}$), −0.84 eV per interfacial atom or −30.2 eV total (corresponding to a value of $S^m E_{adh} = -0.38$ eV Å$^{-2}$,), indicates that the onset temperature of PMC should be ≈427 °C (700 K), just below the lowest temperature at which particle migration was experimentally observed (500 °C).

While there is an increase in energy ($E_{el} = 18.8$ eV, or 0.31 eV per interfacial atom) as a result of nanoparticle elongation on the support, this increase is ≈50% lower than the corresponding change for the elongation of an isolated particle ($E_{el} = 33.3$ or 0.55 eV per interfacial atom, Supplementary Fig. 13). Oxygen from the support is observed to displace towards the Ni nanoparticle at the interface, suggesting that the particle wettability is closely tied to oxygen at the surface of the strontium titanate support. Comprised of a non-noble metal, the Ni nanoparticles may be stabilized through bonding with oxygen at the surface, and experimental evidence for this has recently been shown by Weber et al. [11]. The Ni−O bonds in Fig. 4g−i were sufficiently strong

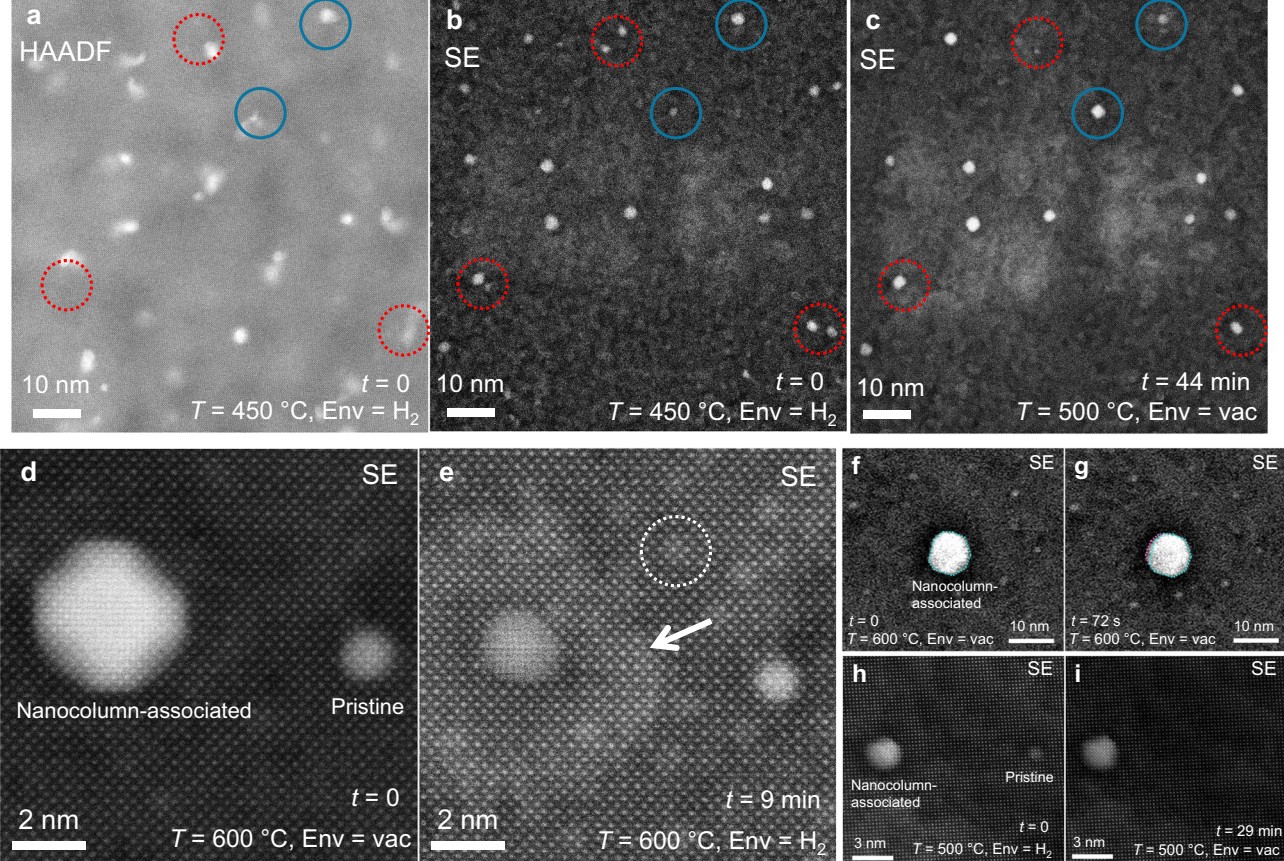

**Fig. 5 | Ostwald ripening and particle redissolution at atomic resolution.**
**a**, **b** HAADF and SE images of a region with several nanocolumn-associated and pristine nanoparticles, and **c** an SE image of the same region after 44 min of coarsening. The circles in (**a**–**c**) indicate regions where particles have either shrunk or grown through OR (nanocolumn-associated particles circled in solid blue lines, pristine in red dashes). **d**–**f** Show the evolution of a region with one of each particle type at 600 °C. After exposure to hydrogen, **e** shows that Ni has diffused away from the larger particle. Small islands of bright contrast are visible (circled in white), and a large region of bright contrast (white arrows) surrounds the nanocolumn-associated particle after shrinking, hypothesized to be surface regions of higher Ni content. Additional processes seen include the growth of a nanocolumn-associated particle (**f**, **g**) and complete redissolution of a pristine particle (**h**, **i**). The areas in (**d**–**g**) were live-imaged continuously (see Supplementary Movies 4, 5, and 7).

to displace oxygen atoms from the support surface towards the Ni nanoparticle. DFT results thus suggest that the particle wettability is closely tied to oxygen at the surface of the strontium titanate support. The wetting model of diffusion present in Fig. 4d–f includes a thermodynamic driving force for diffusion, which is not present in a random-walk diffusion process. However, variations in chemistry (such as changes in oxygen vacancy concentration or cation stoichiometry) or thermal fluctuations at the surface could create the local variations in capillary forces to induce a series of particle movements to form the random-walk kinetics observed in the experiment.

**Dynamics of exsolved particles: dissolution and ripening**
Simultaneous to particle migration, Ostwald ripening was occurring over the course of the in-situ experiments, evident as the simultaneous dissolution of certain particles and growth of others. The evidence for OR is shown at low resolution in Fig. 5a–c, where multiple locations of particle growth/shrinking are noted after 44 min of heating the sample in the range of 450–500 °C. Both nanocolumn-associated and pristine particles are observed to ripen over the course of the experiment. The OR-correlated nanoparticle dissolution process was observed at high resolution in Fig. 5d, e: after an initial exsolution in hydrogen, the region was equilibrated in vacuum at 600 °C (Fig. 5d), then upon hydrogen reintroduction the large particle decreased in size significantly after 9 min (Fig. 5e). Refer to Supplementary Movies 4 and 5 to observe the dissolution processes in Fig. 5c–e.

After the OR-correlated dissolution, there are two contrast features present in Fig. 5e, f: a region of bright contrast surrounds the nanocolumn-associated particle which has partially dissolved (indicated by the white arrow), and there are several 'islands' of bright contrast (circled in white), all which have a diameter of ≈0.5 nm. It has been demonstrated in the literature that SE emission is dependent on the electronic structure of the sample surface[49] and that SE imaging in the STEM is highly sensitive to atomic-scale surface topography[29,50]. It is hypothesized that the regions of brighter contrast correspond to surface regions with an increased Ni content, resulting either from Ni-clustering on the surface or a modified space charge region from local Ni-doping, both of which could increase SE emission from the surface. Supporting this assertion is that the brighter contrast only becomes apparent after significant Ni redissolution has occurred, and is centered on the particle, which has shrunk. With similar argumentation, the small islands of bright contrast (circled in white) may be mobile Ni surface clusters that are active as transport species during Ostwald ripening. The ability to observe near-surface regions with altered chemistry is an added benefit of the use of SE imaging for in-situ experiments; however, the interpretation of SE contrast is complex. In addition to indicating a change in chemistry or topology, a variation in SE contrast may signify a change in electronic structure[51]. Furthermore, up to 20% of the SE signal could be arising from backscattered electrons, as demonstrated by Inada et al.[36]. Similar small contrast features were also observed near other nanoparticles in the system, in both

vacuum and hydrogen environments (Supplementary Fig. 14). Sudden particle growth events were observed, as shown in Fig. 5f, g, where one side of the particle noticeably increases in size after 72 s. A sudden increase in particle size localized to one side of the particle is consistent with a nucleation-barrier inhibited OR-like process, as would be predicted for the ripening of faceted nanoparticles[22]. Full particle dissolution was also observed (Fig. 5h, i), consistent with the expectation that smaller particles will eventually disappear due to the ripening process.

Overall, Ostwald ripening is the dominant coarsening mechanism under the conditions that were used for the current study. In the region of Fig. 3f, g, ≈20% of the pristine nanoparticles were observed to be highly mobile. Despite particle mobility, no cases of particle coalescence could be identified. In certain cases, particle mobility was only observed after particles had first shrunk by Ostwald ripening (Supplementary Movie 5), suggesting that the mechanisms are linked. While here particle mobility was observed in smaller particles, a nucleation-barrier inhibited Ostwald ripening process would suggest that at a critical point, PMC will become dominant due to a decrease in the driving force for ripening. In support of this assertion, prior literature on exsolution-active systems has suggested that PMC is the prevalent coarsening mechanism at larger particle sizes[11,52].

### Related phenomena in exsolution catalysts

The in-situ exsolution experiments presented in this work demonstrate the formation of two distinct exsolved particle populations: those that form above a heteroepitaxial embedded nanocolumn, and those that form in the absence of any nanostructures in the film. The application of simultaneous SE and HAADF imaging during in-situ measurements allows for the differentiation in the behavior of nanocolumn-associated and pristine particles. Experimental evidence suggests that the nanocolumn-associated particles are pinned by the underlying defects, which is attributable to the bonding between the nanocolumns and the Ni particles. Given the reducing conditions during exsolution, it is expected that the full $NiO_x$ nanocolumn is reduced during the exsolution reaction, forming a Ni particle which is bonded to a Ni nanocolumn reaching deep into the substrate. It should be noted that this effect is distinctly different from what has been described in the literature as 'socketing'[10], which has been shown to form as a result of the wetting behavior of nanoparticles on the support oxide[52]. There are additional factors that could impact the metal-support interaction, such as the presence of strain in the lattice surrounding the $NiO_x$ nanocolumns (Supplementary Fig. 8b). The nanocolumn-associated particles are observed to be more stable than the pristine particles, remaining with the same particle density over the in-situ experiment. The bonding with subsurface Ni columns prevents movement of Ni particles, limiting any coarsening of nanocolumn-associated particles by PMC. However, the dissolution of particles and the diffusion of Ni adatoms on the strontium titanate surface (the mechanism for OR) are still present. The OR process is instead dependent on the particle sizes[22], which tend to be larger (therefore more stable) for the nanocolumn-associated particles.

Prior in-situ TEM exsolution studies have focused on characterization of the initial stages of nanoparticle formation[27,28,53]. Han et al.[27] have shown that anti-phase boundaries are preferred sites for exsolution, similarly to how exsolution on $NiO_x$ nanocolumns is observed in the current work. The present study directly demonstrates that nanoparticles that exsolve above defects in the support are stabilized relative to other particles. It is hypothesized that the stability of nanoparticles could also be enhanced in the case of other crystallographic defects, providing a potential route for enhancing the stability of exsolution-active catalysts further. In total, the presence of the $NiO_x$ nanocolumns allows for the production of two populations of Ni nanoparticles, which show distinctly different sizes and coarsening behaviors over the course of the in-situ experiments, presenting an opportunity for future design of multifunctional catalyst systems. This possibility has been partially validated thanks to recent work from Xuan et al.[54].

The utilization of in-situ atomic-resolution SE imaging has allowed for insights into the two fundamental mechanisms of catalytic surface area loss (Ostwald ripening and particle migration and coalescence), which are applicable across all oxide-supported metal catalyst materials. The visualization of particles on a flat surface during in-situ experiments enables quantitative analyses of particle migration in the STEM. Particle movement is observed to be consistent with a random-walk model; however, particle shape changes during movement indicate that particle motion is controlled by wetting physics. The suggestion that particle mobility is analogous to classical wetting physics opens the possibility of utilizing surface-engineered supports to closely control particle coarsening behaviors. In the case of Ostwald ripening, particle dissolution is observed with atomic-resolution secondary electron imaging. Changes in surface contrast after particle dissolution suggest regions of increased surface Ni content, pointing towards the presence of mobile surface species that are active in particle ripening. The results provide evidence of local surface modification of the strontium titanate support in the regions surrounding Ni particles after the OR-associated dissolution process has occurred. Overall, Ostwald ripening was the most evident form of nanoparticle coarsening during the experiment, generally fitting the presumption that exsolution promotes the formation of nanoparticles that are in a stable relationship with the oxide support. In fact, while particle mobility was commonly observed, no coalescence events were directly visualized during the experiments. The thermal stability of nanoparticles formed through exsolution is shown to be complex, with both Ostwald ripening and particle migration occurring simultaneously. Insights into particle migration and Ostwald ripening are applicable not only to exsolution-active systems, but to the coarsening processes in oxide-supported metal catalysts more generally.

## Methods

Epitaxial ([001] oriented) Ni-doped strontium titanate (STNi, with the composition: $SrTi_{0.95}Ni_{0.05}O_{3-\delta}$) thin films were deposited by RHEED-PLD (reflection high-energy electron diffraction-controlled pulsed laser deposition) onto a $Nb:SrTiO_3$ single-crystal substrate. An excimer laser with a wavelength of $\lambda = 248$ nm was used for the deposition, with a repetition rate of $f = 5$ Hz and a laser fluence of $F = 1.14$ J cm$^{-2}$. The substrate temperature during deposition was $T = 650$ °C, and the oxygen partial pressure was $p(O_2) = 0.108$ mbar. The films were grown to a thickness of 150 nm (≈384 unit cells).

Samples for both in-situ and ex-situ STEM were prepared with a Thermo Fisher Scientific Helios NanoLab 460F1 FIB-SEM[55], with a final milling energy of 2 keV. Plan-view liftouts were prepared using a similar method to what has been shown by Palisaitas[56]. For ex-situ STEM, liftouts were polished using a Fischione 1040 Nanomill, with a final milling energy of 500 eV. Ex-situ STEM characterization was done with either a Thermo Fisher Scientific Titan G2 80–200 CREWLEY[57] (including HAADF imaging, EDS mapping, and integrated differential phase contrast (iDPC) imaging) or a Thermo Fisher Scientific Spectra 300 S/TEM (including HAADF imaging and electron energy loss spectroscopy (EELS) mapping). EDS scans were acquired with a Super-X EDX system (G2), and EELS acquisition was done with a Gatan Continuum 1066 energy filter (GIF), with a dispersion of 0.05 eV. iDPC images were collected using a four-quadrant dark-field detector, with a collection angle of ≈6–35 mrad.

To prepare lamellae for in-situ analysis, a plan-view liftout procedure was used, which is summarized schematically in Supplementary Fig. 10. A plan-view liftout was taken via FIB-SEM from the STNi thin film, and thinned as typically done for an ex-situ analysis. The plan-view liftout was then tilted 90° and transferred onto a MEMS chip, with care taken to ensure it lay flat on the chip (i.e., close to the zone-axis).

After MEMS-chip attachment, the top surface of the lamella was polished with a final energy of 2 kV to guarantee a clean, near-atomically flat surface[58,59]. After ion milling at 2 kV, a thin (≈1–2 nm) amorphized layer is expected on the surface, but evidence has shown that short annealing times at 400 °C result in recrystallization of amorphous strontium titanate thin films[60], so the lamella surface is expected to be fully crystalline during the in-situ heating experiments. This is supported by imaging of the lamella edge after heating, which shows a fully crystalline surface (Supplementary Fig. 15). In-situ STEM experiments were carried out with a Hitachi High-Technologies HF5000 environmental S/TEM, equipped with a secondary electron detector. Pure hydrogen was utilized for in-situ experiments, with gas pressures in the column of ≈1 Pa. Experiments were initially conducted in hydrogen to induce the exsolution reaction, then ≈100 min after exsolution, hydrogen was shut off to limit exposure of the cold field emission gun to high pressures. During the late stages of heating, hydrogen was reintroduced for short periods when imaging at high resolution to track evolution in particle faceting due to a change in atmosphere. All imaging was done at 200 kV. Beam currents during in-situ measurements were in the range of 80–200 pA, depending on the experiment. Particle size measurements were done by hand, with an average of 46 particles measured for each of the three times in Fig. 2h, i. Errors in particle diameter were calculated using the standard error of the mean. Error bars in particle density measurements were calculated by assuming the same percent error as is in the particle diameter measurements. Particle density measurements were performed on images with a larger than 50 nm × 50 nm field of view. Exsolved Ni volume was estimated by assuming spherical particle shapes. Particle sizes were averaged over two separate in-situ measurements, leading to the x error bars in Fig. 2h, i, which are the standard error of the mean of the measurement time.

Data analysis of spectral datasets (EDS and EELS maps) was done using Hyperspy[61]. EDS maps were filtered with an average filter to improve signal:noise without introducing artifacts. For select high-resolution STEM images, a nonlinear filtering algorithm was applied, which has been developed specifically for denoising HRS/TEM micrographs[62]. Some in-situ images and videos were temporally frame-averaged to improve signal:noise. Mapping of the ellipticity of atom columns in an HRSTEM image was done using Atomap[63]. A 'roundness' parameter was utilized to quantify the roundness of nanoparticles imaged with the SE detector. The parameter compares between the perimeter and the area of the shape, and is described in Eq. (1):

$$\text{Roundness} = \frac{4\pi * A}{P^2} \qquad (1)$$

Where $A$ is the area and $P$ is the perimeter of the 2D image of the particle, both of which were calculated using an automatic thresholding applied to the image. As constructed, a roundness value equivalent to 1 is a circle, and the less round an object is, the lower the roundness value will be (for example, a shape with a roundness value of 0.5 is described as less round than one with a value of 0.8).

Trackpy was used for kinetic analysis of nanoparticle movements[64], using the two stationary particles as markers to remove drift from the particle track. For the particle tracking in Fig. 3a–e, a minimum particle jump distance was applied during particle tracking, taken to be two standard deviations above the average jump distance of the stationary particles (any 'jumping' of the stationary particles was assumed to be a result of experimental noise). With this cutoff, ≈97% of spurious jumps are removed from the particle tracking data, assuming a normal distribution in the distance of noise-induced jumps in the data. Supplementary Fig. 16 shows the drift measurement from a stationary particle (in both x and y dimensions) from which the drift correction was applied. The oscillations in the 'x' curve in Supplementary Fig. 16 are assumed to be noise, and these oscillations were

used to calculate the cutoff for a statistically significant particle jump during tracking of the mobile particle. To track particles at lower resolution (Fig. 3f–h), manual marking of particle positions was used in combination with Trackpy. Mean squared displacement versus lag time was plotted in Fig. 3e, and fit to a power law model in Eq. (2):

$$\Delta r^2 = 4D_{\text{NP}}t^n \qquad (2)$$

Where $\Delta r^2$ is the mean squared displacement, $D_{\text{NP}}$ is the diffusivity of the nanoparticle, $t$ is the lag time, and $n$ is the power law exponent. Activation energy for particle motion was calculated from the diffusivities of the particle moving at 500 °C (0.4 Å² s⁻¹) and the average diffusivity of the ten mobile particles at 700 °C (3.4 Å² s⁻¹), assuming the diffusivity fits the following Arrhenius relationship as in Eq. (3):

$$D_{\text{NP}} = D_0 e^{\frac{-E_A}{k_B T}} \qquad (3)$$

Where $D_0$ is a constant, $E_A$ is the activation energy, $T$ is the temperature, and $k_B$ is the Boltzmann constant. Particle shape tracking in Fig. 4a–c was done by applying a local Otsu algorithm for automatic thresholding, through the scikit-image Python package[65]. To facilitate automatic thresholding, a Gaussian blur was first applied to the data.

## Simulation methods

**Thermodynamics of STO and STNi.** To confirm which surface termination is energetically preferred for the (001) surface of SrTiO₃ (STO), DFT calculations were performed on different surface terminations using a standard slab model. For the (001) surface of such a cubic perovskite, the SrTiO₃ slab can be considered to be alternating layers of SrO and TiO₂. As such, it is expected that one of these will be the preferred surface. However, these terminations with excess oxygen atoms on the surface (akin to the opposite metal oxide termination, with the metal atom removed) were also considered. Slabs are constructed by repeating the primitive unit cell of SrTiO₃ along the c direction by an integer number $n$, and introducing a vacuum gap. This would result in two surfaces, one of composition SrO and the other of composition TiO₂. To avoid any polarization that might occur across the slab/vacuum, an additional layer of one oxide is introduced to have a symmetric slab. The composition of the slab is then either [SrTiO₃]$_n$-TiO₂ or [SrTiO₃]$_n$-SrO, for integer $n$. We consider integers $n = 1,…,6$. A final vacuum gap of 15 Å is used.

The surface formation energy is calculated using Eq. (4):

$$E_{\text{form}} = \frac{E_{\text{Slab}} - \left[nE_{\text{SrTiO}_3} + E_{\text{SrO/TiO}_2}\right]}{2N} \qquad (4)$$

Here, $E_A$ gives the energy of system A, $n$ is the number of complete SrTiO₃ formula units, $N$ is the total number of atoms in the slab, and the factor of 2 is included to account for the fact that there are two surfaces formed. For surfaces with additional oxygen, an additional term for the extra $N_O$ oxygen atoms is included in the square brackets.

$$E_{\text{form}} = \frac{E_{\text{Slab}} - \left[nE_{\text{SrTiO}_3} + E_{\text{SrO/TiO}_2} + \frac{N_O}{2}E_{O_2}\right]}{2N} \qquad (5)$$

The results of the above equations are presented in Supplementary Fig. 18 for different surface terminations and different thicknesses of slabs. From this, it is clear to see that the surfaces without any additional oxygen (i.e., the SrO and TiO₂ terminations) have the lowest formation energy and are thus the easiest to form. Of these, the SrO surface is the most favorable, which agrees with the results presented in Supplementary Fig. 15. However, it should be noted that the formation energy calculations were deemed to be inconclusive on which of the SrO and TiO₂ terminations are favored, fitting with experimental

work which has shown that mixed termination is present in the as-grown films[16].

As a compromise between chemical accuracy and computational cost, in the following results, we utilize a slab corresponding to $n = 2$, i.e., a slab which is two full unit-cells thick, with an additional SrO layer to ensure there is no polarization across the slab. The difference in energy between a slab of thickness $n = 2$ and $n = 6$ is less than 0.06 eV per formula unit.

**STO-supported Ni nanoparticles.** DFT modeling of unsupported and STO-supported Ni nanoparticles was performed on static particle arrangements, with the aim of closely reproducing the experimental systems in Fig. 4a–c. Two different models were built for this purpose: one of a 'normal' pristine particle on an SrTiO₃ substrate, and one of an 'elongated' particle. The 'normal' particle orientation and facets were chosen to fit as closely to what was observed in high-resolution imaging of other particles (see Fig. 2b, Supplementary Fig. 12). The size of the particle was chosen to fit what was measured in Fig. 4a, the particle before elongation. The 'elongated' particle was assumed to maintain the same orientation relationship with the substrate and the same total number of atoms as the 'normal' particle model (409 Ni atoms in total). The particles were modeled both attached to the support and isolated in vacuum, resulting in four different particle models (Supplementary Fig. 13). Refer to Supplementary Fig. 17 for top-down images of the SrTiO₃-supported crystal models made for DFT simulations (cf. experimental pictures in Figs. 2b, 4a, b). The SrTiO₃ substrate was modeled as a 2-layer slab with an 11 × 11 surface cell and (001) surface orientation (1452 atoms in total), which was constructed from the DFT-relaxed cubic STO bulk cell ($a = 3.88$ Å). SrO terminating layers were chosen, as was observed at the edge of the experimental sample during heating (Supplementary Fig. 15). The simulation box for isolated Ni nanoparticles was chosen with identical dimensions as for the STO-supported systems. DFT simulations of the isolated and supported nanoparticle systems were performed using the VASP software package[66] with the PAW (projector augmented wave) pseudopotential method[67] and the PBE (Perdew-Burke-Ernzerhof) exchange-correlation functional[68]. A plane-wave-basis cutoff energy of 400 eV was used, and Fermi-Dirac smearing with $kT = 0.05$ eV was applied. Due to the large simulation box, $K$-space sampling was restricted to the single $\Gamma$-point. Calculations were performed using the non-spin-polarized DFT method. The negligible influence of spin polarization was confirmed for the individual systems of STO support and isolated Ni nanoparticles. For all systems, atom positions were relaxed until all forces were smaller than 0.05 eV Å$^{-1}$. The D3(BJ) method[69] was used for vdW correction.

Wetting simulations of a liquid droplet on a heterogeneous support were done (Fig. 4d–f) using the evolution equation described by Shwartz and Eley[70], assuming a flat surface, volume conservation, and no impact from gravity. Some figures in the manuscript were made utilizing the matplotlib[71] and cmasher[72] Python packages. Crystal models for figures and DFT modeling were made using Vesta[73].

## Data availability
The data that support the findings of this study are available from Figshare[74] and from the corresponding author upon request.

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

## Acknowledgements

Funding for this work was provided by the Deutsche Forschungsgemeinschaft (DFG) via the Emmy Noether Programme, contract No. Rh 146/1-1, and through projects MA 1280/69-1 and 441718867. D.J. thanks Devan Jennings for assistance in proofreading the manuscript. The authors thank Dr. J. Barthel for discussions related to secondary electron image formation and interpretation. The authors thank the support of Hitachi High-Technologies, in providing access to and support in the operation of the Hitachi HF5000 that was utilized for this work. T.B., C.P., and P.K. contributed to this work within the framework of the Helmholtz Association's program Materials and Technologies for the Energy Transition, Topic 2: Electrochemical Energy Storage. DFT simulations were performed on the JURECA machine in the scope of the project cjiek61.

## Author contributions

Conceptualization and experimental planning: D.J., M.L.W., F.G., and W.R. Experimental data acquisition: D.J., M.L.W., A.M., M.K., H.M., and P.C. Experimental data analysis and interpretation: D.J., M.L.W., A.M., M.H., I.R., J.M., F.G., and W.R. Project supervision: O.G., R.D., J.M., F.G., and W.R. Design and execution of computational results: T.B., D.J., C.P., and P.K. Writing, original draft: D.J. Writing, revision, and editing: all authors.

## Funding

## Competing interests

The authors declare no competing interests.
