## [Transparent Peer Review file · Nature Communications]

Direct atomic-scale investigation of the coarsening mechanisms of exsolved catalytic Ni nanoparticles

Corresponding Author: Dr Dylan Jennings

Version 0:

Reviewer comments:

Reviewer #1

(Remarks to the Author)

This manuscript explores the evolution of Ni nanoparticles generated via exsolution from a spinel containing Ni. Since exsolved particles are considered to have beneficial properties, such as improved thermal stability, it is important to investigate the underlying phenomena. The authors have used advanced EM techniques to address this challenge.

Overall, the work is done using state-of-the-art tools, the manuscript is well written, and the methods and conclusions are sound. The authors prepared bulk samples, then prepared thin sections and mounted them on a MEMS chip for in-situ study in a TEM. They used both HAADF imaging, which looks through the sample and SE imaging, which sees the surface, to examine how the exsolved particles change when heated in a H₂ atmosphere. I particularly liked the microscopy, which shows SE and HAADF imaging at the atomic scale. I have a few comments that the authors can address, which are minor and the work is eminently suitable for publication:

- 1) The SE image in Figure 2a shows atomic scale detail. Is this a reflection of the resolution possible via SE (where the electron generation is thought to be delocalized) or rather is it possible because of the channeling that occurs with this aligned sample? Would such atomic resolution images, showing facets for example, be obtained if the sample was randomly oriented?
- 2) In figure 2 h and i, we see the decrease in number of Ni particles, but why is there no commensurate increase in particle size for the pristine particles. Are you losing Ni to the bulk? Is this consistent with the mass balance? Also, the temperature of this experiment and the atmosphere needs to be mentioned. And this is true for many images in the manuscript where you report the evolution of particles, the temperature and gas phase needs to be mentioned in the figure caption (for example see figure S9)
- 3) The authors show two different behaviors, for the embedded particles and the pristine particles. The embedded particles don't move but the pristine particles appear to show random motion. The movement of these exsolved particles is surprising since they must have formed from Ni diffusing from the bulk. Have they studied the interface between the particle and the support to see why these particles are able to move, even though they are epitaxial to the substrate. I find this rather surprising and indeed not what one would have expected from exsolved particles.
- 4) While the authors state that both OR and PMC is seen, they primarily observed particle migration and no coalescence. This is largely true of such studies in the literature, where the observed random motion does not necessarily result in coalescence. Hence, the coarsening is mostly associated with OR (see ref 1 below).
- 5) With regard to the dissolution, I am surprised that dissolution can occur under reducing conditions. It is normally associated with oxidizing conditions.
- 6) Also, I would like to caution about the interpretation of the bright regions, line 311 after redissolution. Since no elemental analysis was presented, this could also be due to changes in the electronic structure of the support, such as the work function, causing changes in SE emission. We have observed this previously (unpublished work).
- 7) Finally, I do have some concerns about the proposed mechanism for migration, that involves wetting. The authors quote a rather large energy barrier (16 eV) which cannot be overcome at the low temperature used by the authors, 400 C.

Benavidez, A.D., et al., Environmental transmission electron microscopy study of the origins of anomalous particle size distributions in supported metal catalysts. ACS Catalysis, 2012. 2(11): p. 2349-2356.

Reviewer #2

(Remarks to the Author)

In this work, Jennings and co-authors investigated the exsolution of Ni particles and their subsequent migration and coarsening behaviors on a thin film of Ni-doped strontium titanate using in situ STEM. Through simultaneous imaging via HAADF-STEM and SE, they identified two modes of particle exsolution: nanocolumn-associated and pristine. They further discovered that while particles exsolved from the former mode are relatively stable and resistant to motion, those from the latter mode can migrate across the support. This migration follows a random walk model on the film, with variations in particle shape during motion governed by wetting physics. Finally, they demonstrated the occurrence of Ostwald ripening simultaneously with particle motion. Overall, the manuscript is well written and presents some interesting findings. However, major revisions are necessary before I can recommend its publication in Nature Communications.

Here below are my comments:

Caption of Figure 1: There are two instances of “figure c”, while the latter should be labeled as figure d.

It would be beneficial for readers if the authors could provide a brief description of the experimental conditions within insets of figures 2 and 3, similar to what was done for figure 5.

Caption of Figure 2: “All images were collected at 500 °C, either in vacuum (b,d) or an atmosphere of ~1 Pa hydrogen (a,c).”

It is not specified under which conditions figure e was recorded. Additionally, figure c is an FFT image of figure b; however, it is stated that figure c was collected under hydrogen while b was under vacuum. Please make the necessary corrections.

The statement, “In combination with the diffusivity measured at 500 °C, an activation energy for Ni particle diffusion can be estimated to be ~0.6 eV.” requires clarification on how this activation energy was estimated.

The SE images display a significant brightness difference between the Ni NPs and the support. What accounts for the significantly brighter contrast of the Ni NPs compared to the support? Do those SE images also contain signals from backscattered electrons, in addition to secondary electrons?

The influence of the electron beam is not evaluated in this work. What dose rates were used? This should be carefully considered and stated to determine whether the electron beam had any impact on the observed behaviors.

Regarding figures 2h and 2i, how many particles were counted for plotting? Providing this information would be beneficial.

Figures 3b and 3c indicate that, over time, the lattice fringes of the support disappear; was this a result of beam irradiation?

Upon examining figures 3f and 3g, it is observed that after 2 minutes of in situ observation, the topological surfaces, particularly around the Ni NPs, change. It appears that particle migration causes variations in the surfaces. Can authors comments on this?

“The larger jumps match closely with the {100} type crystallographic directions of the support, ...are locked to crystallographic orientations.” I find this assertion questionable based on the video, as the evidence presented is not sufficiently clear.

Additionally, the conclusion appears to be drawn from the observation of a single particle, which is inadequate for a robust analysis.

Similarly, the conclusion regarding PMC, derived from a random-walk diffusion model, seems to rely on tracking the trajectory of one particle as well. To make a conclusive statement, the analysis of a single particle is insufficient.

I recommend that the authors include more experimental details, such as temperature, atmosphere, or vacuum conditions, within the videos instead of merely providing timestamps.

Page 9: It remains unclear why the experiment was conducted initially in hydrogen, followed by a vacuum at elevated temperatures, and then again in hydrogen. A rationale for this sequence of conditions is necessary.

The author stated, “Pure hydrogen was utilized for in situ experiments, with gas pressures in the column in the range of 1-10 Pa” in the Materials and Methods section. However, the manuscript only presents in situ STEM and SE images of samples examined in a vacuum or under an atmosphere of 1 Pa hydrogen (see Fig. 2). Please verify whether data from in situ experiments conducted at a hydrogen atmosphere of 10 Pa is included in this paper. Given that the authors mentioned a different pressure, I am curious whether varying pressures of hydrogen have any impact on particle exsolution or migration behaviors.

What is the role of hydrogen in particle motion? Was particle motion observed under vacuum conditions?

While exsolution predominates in this material system in the presence of hydrogen, particle dissolution was also observed.

What is the driving force or mechanism behind this dissolution behavior in hydrogen?

Page 11: “In the absence of surface curvature, particle movement is consistent with a random walk model”. I will be careful about this statement, as the topological surfaces of the support are not locally flat, as demonstrated in Figures 3f, 3g, S17, S19, and S20.

Version 1:

Reviewer comments:

Reviewer #1

(Remarks to the Author)

The authors have revised the manuscript and addressed most of the reviewer comments satisfactorily. The manuscript is now suitable for publication and will add new insights into the behavior of exsolved Ni particles.

Reviewer #2

(Remarks to the Author)

The authors have done a good job revising their manuscript and addressing my comments. I am pleased to recommend the publication of the revised manuscript in Nature Communications.

Response letter regarding the following manuscript, submitted to the journal *Nature Communications*:

‘Direct Atomic-Scale Investigation of the Coarsening Mechanisms of Exsolved Catalytic Nanoparticles’ *Dylan Jennings, Moritz L. Weber, Ansgar Meise, Tobias Binniger, Conor Price, Moritz Kindelmann, Ivar Reimanis, Hiroaki Matsumoto, Pengfei Cao, Regina Dittman, Piotr Kowalski, Marc Heggen, Olivier Guillon, Joachim Mayer, Felix Gunkel, and Wolfgang Rheinheimer*

We would like to thank the reviewers for the effort which has gone into providing detailed feedback on the manuscript. On the following pages one will find a comprehensive response to each point from both reviewers. For the convenience of the reader, the reviewer comments are repeated, followed by our response in bold, italic lettering. The corresponding changes made in the manuscript are highlighted in red.

In addition to editing of the main text, additional data and analysis has been provided in the supplementary information to address the comments from both reviewers. We have added several supplementary figures which address the main concerns of the reviewers, including those which relate to beam effects, additional kinetic measurements, and others. We also have added a full text section to the supplementary information to address any concerns about the possibility of beam effects influencing the observations. In addition, we have ensured figures are completely labelled with experimental conditions during acquisition.

We fully feel that the manuscript has been improved through the detailed feedback from the reviewers, and we thank them for their time spent reviewing the manuscript.

Sincerely,

Dr. Dylan Jennings

Dr. Moritz L. Weber

Dr. Felix Gunkel

Prof. Dr. Wolfgang Rheinheimer

Reviewer #1 (Remarks to the Author):

This manuscript explores the evolution of Ni nanoparticles generated via exsolution from a spinel containing Ni. Since exsolved particles are considered to have beneficial properties, such as improved thermal stability, it is important to investigate the underlying phenomena. The authors have used advanced EM techniques to address this challenge.

Overall, the work is done using state-of-the-art tools, the manuscript is well written, and the methods and conclusions are sound. The authors prepared bulk samples, then prepared thin sections and mounted them on a MEMS chip for in-situ study in a TEM. They used both HAADF imaging, which looks through the sample and SE imaging, which sees the surface, to examine how the exsolved particles change when heated in a H₂ atmosphere. I particularly liked the microscopy, which shows SE and HAADF imaging at the atomic scale. I have a few comments that the authors can address, which are minor and the work is eminently suitable for publication:

We would like to thank the reviewer for their detailed review and positive opinion of the work. We have worked to address their comments completely. Please refer to each individual comment below for specific responses and for descriptions of changes which have been made to the manuscript.

1) The SE image in Figure 2a shows atomic scale detail. Is this a reflection of the resolution possible via SE (where the electron generation is thought to be delocalized) or rather is it possible because of the channeling that occurs with this aligned sample? Would such atomic resolution images, showing facets for example, be obtained if the sample was randomly oriented?

We are confident that the atomic resolution SE images we present are consistent with the capabilities of SE imaging reported previously for ex-situ analyses. Atomic resolution SE imaging has been demonstrated to be possible by several groups previously¹⁻⁴. The contrast in atomic resolution SE imaging is mainly arising from inner-shell ionization events triggered by the primary, focused, coherent electron probe. The reviewer correctly points out that electrons with a low ionization threshold (often valence band electrons) will contribute to a uniform background signal. The following quote from Brown et al.⁴ describes this effect in detail: “The probability density for wave functions of electrons with low binding energies tends to be delocalized from the atomic positions. They are therefore more affected by the surrounding crystal structure and are often in the valence bands of the crystal. The valence bands produce a diffuse background to secondary electron images and do not provide atomic resolution information.”

Whether atomic resolution SE imaging is possible for a sample with random orientation would be highly dependent on the individual material and conditions. Actually, it has been suggested that the influence of channelling is less than with annular dark field (ADF) imaging. Inada et al³ have studied the influence of sample tilt on SE image contrast, and note that the contrast in ADF imaging drops faster than in SE imaging. This would suggest that SE is less dependent on electron channelling than other imaging modes. However, they still see a significant contrast reduction beyond 2° away from the zone-axis, indicating that atomic resolution SE imaging of randomly oriented samples is generally not feasible. The following statement has been added to the manuscript to elaborate on the SE contrast formation:

Atomic resolution SE imaging at room temperature has been demonstrated to be possible by several groups previously^{29,35-37}. The contrast in atomic resolution SE imaging arises from inner-shell

ionization events triggered by the primary electron beam, and prior literature has indicated that SE contrast is less sensitive to specimen tilt off-zone than other imaging modes³⁶.

2) In figure 2 h and i, we see the decrease in number of Ni particles, but why is there no commensurate increase in particle size for the pristine particles. Are you losing Ni to the bulk? Is this consistent with the mass balance? Also, the temperature of this experiment and the atmosphere needs to be mentioned. And this is true for many images in the manuscript where you report the evolution of particles, the temperature and gas phase needs to be mentioned in the figure caption (for example see figure S9)

The reviewer points out an interesting phenomenon in the particle size evolution of the pristine particles. We argue that the main reason for the decrease in number of pristine particles is mainly a result of these particles being absorbed into the larger, nanocolumn-associated nanoparticles in the system through the typical coarsening process. As discussed in the manuscript, the pristine nanoparticles are more prone to coarsening through both main mechanisms: Ostwald ripening (OR) and particle migration and coalescence (PMC). They are more mobile, as they are not pinned by subsurface defect structures, and they are smaller, meaning they are more prone to redissolution due to the increased chemical potential of Ni at the surface of particles with a smaller diameter. As a result, the majority of the pristine particles redissolve and are incorporated into the larger, nanocolumn-associated particles. Some pristine particles likely also grow over the course of the experiment, accounting for the slight increase in particle size. However, nanoparticles are also likely exsolving during the experiment, which results in the size remaining almost unchanged.

Upon the comment from the reviewer, we have re-evaluated the particle densities and have calculated particle volumes. We have confirmed that particle densities decrease over the course of the experiment. However, the overall exsolved volume increases from the initial to the final measurement time in the experiment (a supplemental figure, Fig. S4, has been added to show this result, see below). The increase in Ni volume suggests that Ni continues to exsolve over the course of the experiment (likely through a combination of precipitation of new particles and growth of existing particles), and that the decrease in particle density can be explained by particle coarsening. The following text has been added on page 4 to address this result:

An approximation of exsolved Ni volume (Fig. S3) shows that the overall Ni volume increases during the experiment, indicating that exsolution occurs simultaneously to nanoparticle coarsening.

Fig. S4:

Figure S4: Complete comparison of exsolution behavior during the in-situ experiment. (a,b) data from Fig. 2, combined with a plot of the total volume of exsolved Ni over the course of the experiment (c). The amount of exsolved Ni increases slightly over the course of the experiment.

The authors would like to thank the reviewer for pointing out the missing temperatures and environments in some of the figure captions, we regret the oversight in the original manuscript. The temperatures and atmospheres in these experiments have been added into all captions in both the manuscript and the supporting information. See below for a list of changes made to address the issues:

Fig. 2 caption:

The data in (h,i) was collected during experiments with temperatures beginning from 400 °C and reaching up to 600 °C, in hydrogen for ~100 mins then in vacuum for ~100 mins. The exact temperature ramps and conditions are displayed in Fig. S5.

Fig S5:

Figure S5: Particle evolution in terms of density (a) and diameter (b) from Figure 2, with experimental temperature ramps included. The two black lines correspond to the temperature ramps of the two separate experiments from which the data was collected. The vertical line indicates the point in which the atmosphere was changed from 1 Pa H₂ to vacuum (within an error of ±6.5 min).

Fig. 3 caption:

Data in (a-c) was collected at 500 °C in vacuum, and data in (f,g) was collected at 700 °C in vacuum.

Fig. 4 caption:

The images in (a-c) were collected at 500 °C in vacuum.

Fig. S19 (now Fig. S11) caption:

The images in (a) were collected at 500 °C in vacuum and those in (b) were collected at 400 °C (pre) and at 500 °C (post), both in hydrogen.

Fig. S20 (now Fig. S12) caption:

All images were collected at 500 °C in vacuum.

Fig. S12 (now Fig. S14) caption:

Experimental conditions: (a,b) 500 °C, vacuum; (c) 450 °C, hydrogen; (d) 500 °C, hydrogen.

Fig. S13 (now Fig. S15) caption:

High angle annular dark field imaging of the sample edge at 300°C in vacuum,...

Fig. S17 (now Fig. S19) caption:

The images in (a,b) were collected at 700 °C in vacuum.

Fig. S18 (now Fig. S21) caption:

The temperature was held constant at 500 °C.

Fig. S20 (now Fig. S23) caption:

Both images were collected at 600 °C in vacuum.

3) The authors show two different behaviors, for the embedded particles and the pristine particles. The embedded particles don't move but the pristine particles appear to show random motion. The movement of these exsolved particles is surprising since they must have formed from Ni diffusing from the bulk. Have they studied the interface between the particle and the support to see why these particles are able to move, even though they are epitaxial to the substrate. I find this rather surprising and indeed not what one would have expected from exsolved particles.

The referee makes an important point here. We believe that in this system, the nanoparticles do not show the typically reported 'socketing' type behavior, which is often the explanation for why exsolved nanoparticles are more stable. In prior work, we have examined cross sections of particles which show no significant socketing behavior⁵. However, as the reviewer indicates, the particles clearly nucleate with a preferred orientation relationship to the support oxide. It is in fact significant to point out that just because the nanoparticles form through exsolution, does not mean they are immune to the typical thermodynamically-driven coarsening mechanisms (and this is one of the points we wanted to make in this work, as the coarsening behavior of exsolution active systems is often dismissed). This of course includes nanoparticle migration, but also nanoparticle dissolution through an Ostwald ripening-type process. We would acknowledge that the migration phenomenon could be accelerated for nanoparticles which are not formed by exsolution, which we have not explored in this work. To our knowledge, a direct comparison of the nanoparticle mobilities between particles formed through different methods has not been performed in the literature, and is a potential area of interest for further study.

4) While the authors state that both OR and PMC is seen, they primarily observed particle migration and no coalescence. This is largely true of such studies in the literature, where the observed random motion does not necessarily result in coalescence. Hence, the coarsening is mostly associated with OR (see ref 1 below).

We thank the referee for the comment, and agree with them on their assessment of OR being the dominant mechanism for coarsening in the conditions studied. On page 9, we have written

Overall, Ostwald Ripening is the dominant coarsening mechanism under the conditions which were used for the current study. In the region of Fig. 3(f,g), ~20% of the pristine nanoparticles were observed to be highly mobile. Despite particle mobility, no cases of particle coalescence could be identified.

While no coalescence events were observed, prior ex-situ work on this material system suggests that nanoparticle coalescence plays a large role in particle growth, particularly over longer time scales⁵. We would also argue that the presence of particle migration strongly implies that particle coalescence events will also occur, though this proved difficult to identify experimentally. Additional emphasis has been added to the manuscript, with the following text inserted into the discussion on page 11:

In fact, while particle mobility was commonly observed, no coalescence events were directly visualized during the experiments.

5) With regard to the dissolution, I am surprised that dissolution can occur under reducing conditions. It is normally associated with oxidizing conditions.

We agree that the observation of NP dissolution is not typically expected for exsolution-active systems. Importantly, we attempt here to make a distinction between the associated driving forces. The first, as the referee refers to here, would be the dissolution of exsolved nanoparticles due to reoxidation, moving Ni back into the perovskite structure and reforming a solid solution. We agree that this is not probable under the current conditions. However, we do have a driving force for NP redissolution in the form of the OR coarsening process. Smaller particles have a higher Ni chemical potential at the surface, meaning that Ni is more easily moved into the perovskite or onto the surface as adatomic species, as compared to larger particles. By statistical thermodynamic principles, the result is a concentration gradient of adatomic species which flows from small particles towards larger particles, and eventually results in the complete dissolution of some nanoparticles. This process is expected to occur even in reducing conditions. Both OR and a reoxidation process are thermally activated, so particles are expected to be stable at low temperature.

6) Also, I would like to caution about the interpretation of the bright regions, line 311 after redissolution. Since no elemental analysis was presented, this could also be due to changes in the electronic structure of the support, such as the work function, causing changes in SE emission. We have observed this previously (unpublished work).

We would like to thank the reviewer for the comment, and we agree completely that there is not enough evidence currently that the bright regions in the SE imaging post redissolution are Ni clusters. We are of the opinion that they likely correspond to either (1) Ni which is on the surface of the strontium titanate, or (2) Ni which is in the strontium titanate lattice near the surface, which would then result in a change in the local electronic properties and give rise to the change in contrast. We agree that other electronic structure-related properties (such as conductivity and work function) can have an effect⁶. We have changed the text in this section (page 9) to clarify that the mechanism for the contrast change is as-of-yet inconclusive. The text now reads as follows:

The ability to observe near-surface regions with altered chemistry is an added benefit of the use of SE imaging for *in situ* experiments, however the interpretation of SE contrast is complex. In addition to indicating a change in chemistry or topology, a variation in SE contrast may signify a change in electronic structure⁵¹.

7) Finally, I do have some concerns about the proposed mechanism for migration, that involves wetting. The authors quote a rather large energy barrier (16 eV) which cannot be overcome at the low temperature used by the authors, 400 C.

We thank the reviewer for bringing up their concern about the mechanism for migration. We have presented a value of 18.8 eV as an energy cost to the elongation which was experimentally observed. To calculate this value, we compare the energies of the two experimentally observed particle shapes, one which is the starting point (Fig. 4(g)), and one which is the elongated intermediate shape (Fig. 4(h)). Importantly, we are able to show that the particle elongation is facilitated by the particle's bonding to the support – the same elongation of an identically shaped, unsupported Ni particle has an energy cost of 33.3 eV (Fig. S13). Given the facilitation of the elongation by the support, we confirm the importance of the support-particle bonding on the particle migration mechanism. In that sense, we agree that the 18.8 eV value should not be seen as a kinetic barrier.

Based the work of Hu and Li⁷, an additional route would be to consider an energy increase per interfacial atom. In this case, we would expect an energy increase of 0.31 eV/atom for the particle elongation. To limit confusion, the text in this section has been changed to present both values, and to avoid using the terms 'cost' or 'barrier'. The text now reads as follows:

While there is an increase in energy ($E_{el} = 18.8$ eV, or 0.31 eV/interfacial atom) as a result of nanoparticle elongation on the support, this increase is ~50% lower than the corresponding change for the elongation of an isolated particle ($E_{el} = 33.3$ eV, or 0.55 eV/interfacial atom, Fig. S13).

The relevant section of the caption of Fig. 4 has been changed to the following:

(g,h) Density functional theory (DFT) modelling of a static particle in the 'initial' (g) and 'elongated' (h) states. DFT modelling indicates that the increase in energy corresponding to particle elongation (E_{el}) is lowered by its interaction with the support.

To expound further, we have utilized the value of $S_m E_{adh}$ (-0.38 eV/Å²) as a judgement of expected PMC onset, which is a measure of the adhesion energy normalized by the interfacial area. This value has been established as a thermodynamic descriptor for the onset temperature of PMC by Hu and Li⁷, and fits nicely with what we observe during the experiment; that is, an expected onset temperature of ~430 °C (the lowest temperature we observed clear particle migration was at 500 °C). This temperature is also in the range of the Tammann temperature of Ni, at which particle behavior is expected to become similar to that of a liquid droplet. We have altered the text on page 7 to further clarify the meaning of the $S_m E_{adh}$ descriptor. It now reads as follows:

Hu and Li⁷ have developed a thermodynamic descriptor for the expected onset temperature of PMC, based on an interfacial area-normalized adhesion energy, defined as adhesion energy (E_{adh}) multiplied by the number of atoms per interfacial area (S^m). The adhesion energy of the symmetrical particle (E_{adh} , -0.84 eV per interfacial atom or -30.2 eV total (corresponding to a value of $S^m E_{adh} = -0.38$ eV/Å²), indicates that the onset temperature of PMC should be around ~427 °C (700 K), just below the lowest temperature at which particle migration was experimentally observed (500 °C).

Reviewer #2 (Remarks to the Author):

In this work, Jennings and co-authors investigated the exsolution of Ni particles and their subsequent migration and coarsening behaviors on a thin film of Ni-doped strontium titanate using in situ STEM. Through simultaneous imaging via HAADF-STEM and SE, they identified two modes of particle exsolution: nanocolumn-associated and pristine. They further discovered that while particles exsolved from the former mode are relatively stable and resistant to motion, those from the latter mode can migrate across the support. This migration follows a random walk model on the film, with variations in particle shape during motion governed by wetting physics. Finally, they demonstrated the occurrence of Ostwald ripening simultaneously with particle motion. Overall, the manuscript is well written and presents some interesting findings. However, major revisions are necessary before I can recommend its publication in Nature Communications.

We thank the reviewer for their thorough review, and for bringing up several salient points about the article which should be adjusted before publication. We feel that we have corrected the manuscript to fully address the concerns of the reviewer, through changes in manuscript text and figures, the addition of several supplementary figures, and the addition of a new section in the supplemental text. The comments of reviewer 2 have been numbered for convenience.

Here below are my comments:

1) Caption of Figure 1: There are two instances of “figure c”, while the latter should be labeled as figure d.

This error has been fixed.

2) It would be beneficial for readers if the authors could provide a brief description of the experimental conditions within insets of figures 2 and 3, similar to what was done for figure 5.

Upon the suggestion of the reviewer we have adjusted these figures to include conditions as overlays in the images. The new figures are shown below:

Fig. 2:

Fig. 3:

3) Caption of Figure 2: “All images were collected at 500 °C, either in vacuum (b,d) or an atmosphere of ~1 Pa hydrogen (a,c).” It is not specified under which conditions figure e was recorded. Additionally, figure c is an FFT image of figure b; however, it is stated that figure c was collected under hydrogen while b was under vacuum. Please make the necessary corrections.

We thank the reviewer for pointing out this inconsistency in Figure 2. We have fixed the typo, by changing (a,c) to (a,e). Based on the prior comment, the conditions have been added in the images as overlays (see the new figure in comment #2 from reviewer #2).

4) The statement, “In combination with the diffusivity measured at 500 °C, an activation energy for Ni particle diffusion can be estimated to be ~0.6 eV.” requires clarification on how this activation energy was estimated.

The activation energy was calculated by assuming an Arrhenius relationship in the nanoparticle diffusion behavior. This is explained in the experimental details, as follows:

‘Activation energy for particle motion was calculated from the diffusivities of the particle moving at 500 °C (0.4 Å²/s) and the average diffusivity of the ten mobile particles at 700 °C (3.4 Å²/s), assuming the diffusivity fits the following Arrhenius relationship:

$$D_{np} = D_0 e^{\frac{-E_A}{RT}}$$

Equation S3

Where D_0 is a constant, E_A is the activation energy, T is the temperature, and R is the gas constant.'

To clarify further and to acknowledge that limited data was taken on temperature-dependence, the text on page 5 of the main manuscript has been adjusted to the following:

While limited data was taken to evaluate the temperature-dependence, comparison of the diffusivity at 500°C and 700°C would reflect an activation energy of ~0.6 eV, by assuming that the nanoparticle diffusion process follows a typical Arrhenius relationship (Eq. S3).

5) The SE images display a significant brightness difference between the Ni NPs and the support. What accounts for the significantly brighter contrast of the Ni NPs compared to the support? Do those SE images also contain signals from backscattered electrons, in addition to secondary electrons?

We expect that the difference in secondary electron signal between the NPs and the support to be a result of a variety of influences, and it is not completely clear what is the major cause. The SE emission/detection process is dependent on several factors, including surface orientation, topology, and material⁶. While not as heavily dependent on atomic mass as backscattered electrons, material differences (such as comparing between insulators and metals) are known to influence SE emission⁶. This has also been demonstrated to be true at atomic resolution, as Brown et al.⁴ write: 'The valence bands produce a diffuse background to secondary electron images and do not provide atomic resolution information.' Therefore, one expects there to be a higher background signal for the metallic nanoparticles as compared to the insulating strontium titanate support. Additionally, we must emphasize that the image collection was optimized for the full contrast range, so we do not at this point have quantitative analysis of differences in secondary electron emission between the nanoparticles and the support. We hypothesize that the contrast difference arises from a combination of electronic structure (i.e. metallic vs. semiconducting) and the raised topology of the particle contributing to increased signal.

We thank the reviewer for correctly pointing out that the SE signal may contain some backscattered component. Inada et al. have analyzed SE imaging quantitatively, and have shown that up to 20% of the signal reaching the SE detector could arise from backscattered electrons³. To reflect this possibility, we have expanded the discussion on SE signal on page 9, which now reads as follows:

The ability to observe near-surface regions with altered chemistry is an added benefit of the use of SE imaging for *in situ* experiments, however the interpretation of SE contrast is complex. In addition to indicating a change in chemistry or topology, a variation in SE contrast may signify a change in electronic structure. Furthermore, up to 20% of the SE signal could be arising from backscattered electrons, as demonstrated by Inada et al³⁶.

6) The influence of the electron beam is not evaluated in this work. What dose rates were used? This should be carefully considered and stated to determine whether the electron beam had any impact on the observed behaviors.

We thank the reviewer for pointing out the influence of the electron beam, which we agree is always a concern with any in-situ electron microscopy study, and has been reported for STO-related studies in the past⁸. We acknowledge and regret that this concern was not adequately addressed in the initial manuscript. We have added dose rates to the supplementary information, which are in the range of $350 \text{ e}/(\text{Å}^2\text{s})$ to $8.9 \cdot 10^4 \text{ e}/(\text{Å}^2\text{s})$. We are convinced that our results are not significantly influenced by the electron beam based on the following points:

- (1) Importantly, we observe similar dynamics of particle migration and Ostwald ripening at relatively low and relatively high dose rates. We observe particle migration and Ostwald ripening phenomena at dose rates of $\sim 350 \text{ e}/(\text{Å}^2\text{s})$ (Video S6 and S7), $\sim 5 \cdot 10^3 \text{ e}/(\text{Å}^2\text{s})$ (Videos S1 and S3), and $\sim 8.9 \cdot 10^4 \text{ e}/(\text{Å}^2\text{s})$ (Videos S4 and S5). The observed dynamics are qualitatively and quantitatively similar at all three dose rates, strongly implying that changing the electron dosage within these ranges is not significantly influencing the observed behaviors.*
- (2) All of the nanoscale features/dynamics which we observe we see similar evidence for in regions where we have not live imaged during the experiment. In Figure S14, we show contrast features on the surface, consistent with the results in Figure 5(d,e) which are present in regions which were not live imaged. Also in Figure 5(a-c) we directly show a before and after image with regions which were not live imaged in this time period, showing clear locations where nanoparticles have completely redissolved and other locations where particles have grown.*
- (3) The final justification for a lack of significant beam effects is that we observe behaviors which are expected based on ex-situ experiments in the same system, and which fit well to physical models of nanoparticle coarsening. In ex-situ work, clear evidence for particle migration and coarsening in this system (Ni doped strontium titanate thin films) has been presented. By comparison, in a very similar system (Ni and Nb co-doped strontium titanate films), we have observed much more stable particles, consistent with ex-situ results. Were the dynamics to be introduced by the electron beam, then we would expect to observe these behaviors in both systems, which we do not see. Furthermore, our observations of particle mobility fit with the thermodynamically predicted temperature at which Ni mobility would be expected ($\sim 430 \text{ °C}$), following the work by Hu and Li⁷.*

While we are confident that our work does not show a significant beam influence, we must acknowledge that this will not necessarily be the case for other samples or experimental conditions. We have added another supplementary figure which shows the consistency in the in-situ experiments between prior ex-situ literature and the current work. Additionally, we have expanded the supplemental text to include a dedicated section on the evaluation of beam effects in the experiment. The full text and supplemental figures added to the supplementary information are shown here:

Evaluation of Beam Effects during in situ Experiments

Understanding the effect of the electron beam on material behavior during in situ STEM experiments is critical, however no significant beam effect was observed during the experiments presented in this work. For oxide species the presence of the electron beam is known to induce local reduction¹⁴; a local reduction could result in enhanced exsolution around regions which are exposed to the beam, but no evidence of the beam impacting the exsolution process was observed in the current experiment. Several pieces of evidence are presented here to demonstrate that beam effects do not significantly influence the observations in this work.

Firstly, the dynamic processes observed (particle migration and Ostwald Ripening) are qualitatively and quantitatively consistent at varying electron dose rates. Particle migration was observed at relatively low dose ($350 \text{ e}^-/(\text{\AA}^2\text{s})$, video S6), intermediate dose rates ($5.0 \times 10^3 \text{ e}^-/(\text{\AA}^2\text{s})$, videos S1 and S3), and high dose rates ($8.9 \times 10^4 \text{ e}^-/(\text{\AA}^2\text{s})$, videos S4 and S5). Quantification of particle migration at both low and intermediate dose rates shows similar kinetic behavior – that is, behavior which is consistent with a random-walk model. Refer to Fig. S25 for a comparison which shows Brownian motion type kinetics at the two different dose rates. Ostwald ripening processes were also observed at low, intermediate, and high dose rates.

In addition to recording consistent behavior at different dose rates, features similar to those observed during live imaging were also seen in regions of the sample which were not continuously exposed to the electron beam. Figure S14 shows contrast features surrounding non-live-imaged particles which are qualitatively comparable to contrast features observed in Fig. 5(d,e), a region which was continuously imaged. Additionally, Figure 5(a-c) shows a before and after of a region which was not continuously imaged, with clear evidence of the Ostwald Ripening processes that are seen in live imaging (particle redissolution and particle growth).

Finally, the observations made while live imaging the samples are consistent with physical theories for nanoparticle growth kinetics, and fit to observations from ex situ work on the same material system. Nanoparticle exsolution was first observed at $400 \text{ }^\circ\text{C}$, consistent with ex situ work. Furthermore, the temperature at which nanoparticles become mobile on the surface is consistent with expectations, at a temperature of around $500 \text{ }^\circ\text{C}$ (based on work from Hu and Li¹⁵ one expects mobility to appear around $430 \text{ }^\circ\text{C}$). In addition to in situ experiments in the STNi system, comparable in situ experiments were done in the STNNi system (a Ni and Nb co-doped strontium titanate thin film). When comparing the particle evolution between the two material systems, they behave exactly as expected. Nanoparticles in the STNNi system exsolve more slowly, but are also much more stable to coarsening. There were also no observations of particle migration in the STNNi system, supported by ex situ work which posits that particle migration only occurs in the STNi material system. The comparison between this work and the corresponding literature is shown in Figure S24.

Figure S24: Comparison between in situ and ex situ results for two material systems. (a-c) comparison of particle size, density, and exsolved volume for a Ni-doped strontium titanate (STNi, the material of focus in this work) and for a Ni/Nb co-doped system (STNNi, the same material with 5% Nb doping). Atomic force microscopy (AFM) images of the STNNi (d) and STNi (e) show evidence of particle migration only in the STNi sample, consistent with in situ results. Particle density trends in *ex situ* experiments (f) match nicely with what is observed during *in situ* measurements (b). The images in (d-f) are reprinted from Weber et al.¹¹

Figure S25: Power law fits for random walk kinetics for particle in Fig. 3(a-c) (a) and the assembly of particles in Fig. 3(f,g) (b). Both groups of particles fit well to a power law, corresponding to random walk kinetics for particle motion.

7) Regarding figures 2h and 2i, how many particles were counted for plotting? Providing this information would be beneficial.

This information (an average of 46 particles per data point) is included in the experimental details section. For clarity, this information has been added to the caption of figure 2, which reads as:

An average of 46 particles per timestep were evaluated for the data in (h,i).

8) Figures 3b and 3c indicate that, over time, the lattice fringes of the support disappear; was this a result of beam irradiation?

This was not a result of beam irradiation, but instead was due to drift in two-fold astigmatism and focus during the video acquisition, unfortunately a typical effect for the microscope utilized for the experiment. We wished to avoid focussing during video acquisition as the magnification was not high enough to confidently improve the focus during live acquisition. This can be clearly observed in the HAADF signal in video S1 (see below for screenshots from the video). We were able to focus and correct astigmatism in the image during video acquisition in videos S4 and S5 due to the high magnification which allowed us to do this. Though, the downside is that focussing during video acquisition influences the image and has the potential to cause artefacts which then must be

accounted for when analyzing the video. This could be particularly challenging for particle tracking, for example.

Figure: Three screenshots from video S1 showing the slow degradation of the focus condition over the course of the image. Focussing during video acquisition was not done so as to avoid the introduction of artefacts into the particle tracking process.

A comment was added next to the video S1 description in the SI, which now reads as the following:

The focus condition was not changed during the video acquisition, so slight drift in image focus is noticeable over the course of the video.

9) Upon examining figures 3f and 3g, it is observed that after 2 minutes of in situ observation, the topological surfaces, particularly around the Ni NPs, change. It appears that particle migration causes variations in the surfaces. Can authors comment on this?

The authors thank the reviewer for this interesting observation. We acknowledge that the surface of the strontium titanate support could be rearranging during the experiment, though significant movement of surface steps in strontium titanate would be expected at higher temperatures than what we utilize in this work (for example, thermal etching of SrTiO₃ would typically be done at temperatures ≥ 1000 °C). There is also the possibility that, as Ni leaves the B-site of the perovskite, the remaining (now B-site deficient) perovskite may collapse and restructure as a result. However, we have not observed this to be a large effect. In the case of Fig. 3f and 3g, we are not confident enough to say whether the surface is rearranging, due to the relatively low magnification of the images. It remains an open question whether surfaces are affected through the migration of nanoparticles, and is an interesting area of potential further study. We have included an additional supplemental figure (Fig. S20) which shows the STO surface steps at higher resolution, and shows the disappearance of an SrTiO₃ island along with restructuring of STO surface step structures at 700 °C in vacuum, to note that surface changes in the STO are likely occurring during the experiment. See below for the text and figure added to the supplementary information:

The strontium titanate surface facets have been observed to rearrange during the in situ experiments (refer to Fig. S20).

Figure S20: Before (a) and after (b) secondary electron images showing flattening of the SrTiO₃ support at 700 °C in vacuum. The arrow indicates an SrTiO₃ surface island which disappears after 5 minutes. Surface step structures are also observed to evolve in other parts of the image.

10) “The larger jumps match closely with the {100} type crystallographic directions of the support, ...are locked to crystallographic orientations.” I find this assertion questionable based on the video, as the evidence presented is not sufficiently clear. Additionally, the conclusion appears to be drawn from the observation of a single particle, which is inadequate for a robust analysis.

We thank the reviewer for their comment, and upon further consideration have decided to reevaluate this statement. Our intention was to draw an interesting observation from our highest resolution observation of particle migration, in that the larger jumps match closely to the {100} type directions of the support. We agree that the statement ‘are locked to crystallographic orientations’ is not supported by enough evidence in this work. This statement has been softened considerably to be presented more as an observation extracted from the data in Fig. 3, and now reads as follows:

The larger jumps match closely with the {100} type crystallographic directions of the support, suggesting that the jumps that combine to form random walk-type diffusion kinetics may be more favorable along certain crystallographic directions.

11) Similarly, the conclusion regarding PMC, derived from a random-walk diffusion model, seems to rely on tracking the trajectory of one particle as well. To make a conclusive statement, the analysis of a single particle is insufficient.

We agree with the reviewer that the analysis of a single particle is insufficient. We also have the particle tracking from several particles in Fig. 3f,g, which also fits well to random walk kinetics. This graph has been added as an additional supplemental figure (Fig. S25), as support for both

random walk kinetics and for discussion on the effect of electron dose. Fig. S25 is shown above in the response to comment #6 of reviewer 2.

12) I recommend that the authors include more experimental details, such as temperature, atmosphere, or vacuum conditions, within the videos instead of merely providing timestamps.

We thank the reviewer for the suggestion. Upon their recommendation, overlays describing experimental conditions have been added to the supplemental videos.

13) Page 9: It remains unclear why the experiment was conducted initially in hydrogen, followed by a vacuum at elevated temperatures, and then again in hydrogen. A rationale for this sequence of conditions is necessary.

We thank the reviewer for pointing this out, and we regret the confusion introduced by the change in conditions. The initial use of hydrogen was, naturally, to induce the exsolution reaction by creating a highly reducing environment. After exsolution had occurred, we were interested in examining the evolution of the particles, and hydrogen was turned off as it can be damaging for the cold field emission gun to be exposed to high gas pressures for an extended time period. In Figure 5(d,e) we present before and after images looking at regions when we changed back to hydrogen. While it did not end up as the focus of this work, we were interested in observing whether the reintroduction of hydrogen had a significant influence on the morphology of the supported particles. Outside of a slight change in faceting as a result of changes in the anisotropy of the Ni surface energy (see Fig. S21), there was not a large change in particle morphology between the two environments. At the temperatures and gas pressures investigated, the gas environment is reducing enough to maintain the Ni particles as metallic particles, and prevents reoxidation.

Some text has been added to the materials and methods section to address the change in environment over the course of the experiment:

Experiments were initially conducted in hydrogen, to induce the exsolution reaction, then around 100 minutes after exsolution hydrogen was shut off to limit exposure of the cold field emission gun to high pressures. During late stages of heating hydrogen was reintroduced for short periods when imaging at high resolution to track evolution in particle faceting due to a change in atmosphere.

14) The author stated, “Pure hydrogen was utilized for in situ experiments, with gas pressures in the column in the range of 1-10 Pa” in the Materials and Methods section. However, the manuscript only presents in situ STEM and SE images of samples examined in a vacuum or under an atmosphere of 1 Pa hydrogen (see Fig. 2). Please verify whether data from in situ experiments conducted at a hydrogen atmosphere of 10 Pa is included in this paper. Given that the authors mentioned a different pressure, I am curious whether varying pressures of hydrogen have any impact on particle exsolution or migration behaviors.

We thank the reviewer for pointing this out, and regret that we had conflicting information in the original manuscript. In fact, all experiments were done in a hydrogen pressure of ~1 Pa. We wrote 1-10 Pa as the general specification of the instrument (i.e. we could have gone up to 10 Pa of hydrogen). This has been resolved in the manuscript by changing the text in the materials and methods section to the following:

Pure hydrogen was utilized for in situ experiments, with gas pressures in the column around 1 Pa.

The reviewer makes an interesting point about the influence of changing gas pressures. To this point, this is not an aspect which we have studied, though is an interesting area for potential future work. The relative change in pO_2 between 1 Pa and 10 Pa hydrogen would be expected to be very small compared to the relative change in pO_2 between vacuum and hydrogen, but may still have an impact. Recent work from Garcia et al., shows that changes in pressure can have affect both the composition and density of the particles which form⁹, though this is in different pressure ranges from what we have investigated here.

15) What is the role of hydrogen in particle motion? Was particle motion observed under vacuum conditions?

We observed particle migration occurring in both vacuum (Videos S1 and S6) and hydrogen (Video S4) conditions. It is as of now unclear what the role of hydrogen in particle motion is. We hypothesize that hydrogen acts mainly as a reducing agent to drive the exsolution process, but, naturally, a change in the gas environment will change the thermodynamics which may then affect the particle migration process. Perhaps most significantly, the change in gas atmosphere has the potential to influence the surface energies of the Ni particles, changing the relative interfacial energy and therefore the propensity of particles for migration. Additionally, a reduction in the pO_2 will result in an increase in the number of oxygen vacancies at the support surface, and correspondingly change the stability of the nanoparticle-support interface, as described by Weber et al.⁵

We can visualize the influence of hydrogen on surface energies in Figure S21 (reprinted below), which shows a slight change in the faceting behavior of a particle in hydrogen vs. in vacuum. In particular, we see a decrease in the anisotropy of the Ni surface energies when the particles are exposed to vacuum.

Figure S21: The faceting behavior of the particles is observed to change depending on the atmosphere in the microscope. The roundness of the particle (using a comparison between perimeter and area, where Roundness = 1 indicates a circle) is shown to increase in the presence of hydrogen (b), then decrease again when hydrogen is shut off (c). The area was not live imaged in the time between the three images shown here. In addition to the shape change, the pristine particle is observed to disappear between (b) and (c). The temperature was held constant at 500 °C.

16) While exsolution predominates in this material system in the presence of hydrogen, particle

dissolution was also observed. What is the driving force or mechanism behind this dissolution behavior in hydrogen?

We thank the reviewer for bringing up this interesting observation, which we wanted to emphasize as part of this work. While exsolution is the dominant force for nanoparticle precipitation, during nanoparticle coarsening (and Ostwald Ripening in particular) nanoparticle redissolution must be occurring simultaneously. As discussed in the above response to comments from reviewer #1, we have two different mechanisms which could explain the dissolution process which are important to distinguish: the first would be particle redissolution through some reoxidation process – this is not likely to be occurring under these reducing conditions, as is pointed out by the reviewer. However, it has recently become clear that redissolution driven by reoxidation in dilutely doped perovskite exsolution catalysts is quite limited overall ⁹.

The second mechanism would be through Ostwald Ripening. In OR, the high surface curvature of smaller particles result in a higher Ni chemical potential at their surfaces compared to larger particles, resulting in easier formation of adsorbed Ni adatoms (or Ni dopants in the bulk). The end result is a concentration gradient in Ni species from small particles to large particles, providing a diffusion gradient for Ni to move from small to large particles. At some point, the Ni particles will completely dissolve as part of this process.

17) Page 11: “In the absence of surface curvature, particle movement is consistent with a random walk model”. I will be careful about this statement, as the topological surfaces of the support are not locally flat, as demonstrated in Figures 3f, 3g, S17, S19, and S20.

We thank the reviewer for this comment, and in retrospect we agree with them that this statement should be made with additional care. As such, we have removed that part of the statement, leaving the following:

Particle movement is observed to be consistent with a random walk model, however particle shape changes during movement indicate that particle motion is controlled by wetting physics.

References :

1. Ciston, J. *et al.* Surface determination through atomically resolved secondary-electron imaging. *Nat. Commun.* **6**, (2015).
2. Zhu, Y., Inada, H., Nakamura, K. & Wall, J. Imaging single atoms using secondary electrons with an aberration-corrected electron microscope. *Nat. Mater.* **8**, 808–812 (2009).
3. Inada, H. *et al.* Atomic imaging using secondary electrons in a scanning transmission electron microscope: Experimental observations and possible mechanisms. *Ultramicroscopy* **111**, 865–876 (2011).
4. Brown, H. G., D’Alfonso, A. J. & Allen, L. J. Secondary electron imaging at atomic resolution using a focused coherent electron probe. *Phys. Rev. B* **87**, 054102 (2013).
5. Weber, M. L. *et al.* Thermal stability and coalescence dynamics of exsolved metal nanoparticles at charged perovskite surfaces. *Nat. Commun.* (2024) doi:10.1038/s41467-024-54008-4.
6. Seiler, H. Secondary Electron Emission. *Scan. Electron Microsc.* **1982**, (1982).
7. Hu, S. & Li, W.-X. Sabatier principle of metal-support interaction for design of ultrastable metal nanocatalysts. *Science* **374**, 1360–1365 (2021).
8. Jiang, N. Electron beam damage in oxides: a review. *Rep. Prog. Phys.* **79**, 016501 (2016).
9. López-García, A., Remiro-Buenamañana, S., Neagu, D., Carrillo, A. J. & Serra, J. M. Squeezing Out Nanoparticles from Perovskites: Controlling Exsolution with Pressure. *Small* **20**, 2403544 (2024).